# O-mannosylation of misfolded ER proteins promotes ERAD

Leticia Lemus [1✉], Hadar Meyer[2], Ana I Rodríguez-Rosado[1], Maya Schuldiner [2] & Veit Goder [1✉]

## Abstract

**Protein quality control (PQC) in the secretory pathway, a process critically linked to numerous human diseases, begins in the endoplasmic reticulum (ER) and involves ER-associated degradation (ERAD) of terminally misfolded proteins. In this study, we conducted genome-wide screens in baker's yeast (*Saccharomyces cerevisiae*) to investigate the degradation of Gas1\*, a misfolded version of an O-mannosylated, glycosylphosphatidylinositol (GPI)-anchored protein. In combination with detailed biochemical and genetic analyses, these screens revealed an unexpected bifunctionality of the evolutionarily conserved heteromeric enzyme complex Pbn1-Gpi14: while it has been previously recognized as a GPI-mannosyltransferase, we here find that it catalyzes the O-mannosylation of misfolded proteins, thereby promoting their ERAD. This process is particularly relevant for misfolded proteins that lack N-glycans. Our results suggest that protein O-mannosylation constitutes a distinct type of glycan-dependent mechanism for promoting ERAD.**

**Keywords** Protein O-Mannosylation; ERAD; ER Quality Control; Pbn1; Gpi14
**Subject Categories** Membranes & Trafficking; Organelles; Post-translational Modifications & Proteolysis

## Introduction

Folding of proteins in the secretory pathway starts after their translocation across or integration into the membrane of the endoplasmic reticulum (ER) and is monitored by ER protein quality control (ERQC), a process that is essential for cellular homeostasis (Ellgaard and Helenius, 2003; Sitia and Braakman, 2003). ERQC directs non-aggregated terminally misfolded polypeptides to ER-associated protein degradation (ERAD), which involves their retrotranslocation into the cytosol and ubiquitination by ER membrane-embedded protein complexes prior to delivery to the proteasome (Hiller et al, 1996; Werner et al, 1996; Swanson et al, 2001; Denic et al, 2006; Carvalho et al, 2006; Nakatsukasa et al, 2008; Biederer et al, 1997; Jarosch et al, 2002; Stein et al, 2014; Bodnar and Rapoport, 2017).

Most proteins in the secretory pathway undergo N-glycosylation within the ER, playing crucial roles in ERQC and ERAD. Initially, preassembled polyglycan structures are added to site-specific asparagines in luminal protein domains (Helenius and Aebi, 2004; Ruiz-Canada et al, 2009). Subsequent enzymatic glycan trimming is coupled to protein folding, generating N-glycan structures that either promote (re-)binding to lectin-type chaperones for folding or target them for delivery to ERAD (Hammond et al, 1994; Caramelo et al, 2004; Molinari et al, 2004; Quan et al, 2008; Clerc et al, 2009). Although misfolded proteins lacking N-glycans can also be efficiently routed to ERAD (Leto et al, 2019), it remains unclear whether, and by what mechanisms, these substrates are selectively labeled.

Protein O-mannosylation is a second type of glycosylation that occurs inside the ER. Members of the evolutionarily conserved ER-resident protein mannosyltransferases (PMTs/POMTs) attach single mannoses primarily to serines or threonines, with most data coming from studies in yeast (Haselbeck and Tanner, 1983; Lommel and Strahl, 2009; Neubert et al, 2016). Mammalian cells have additional specialized protein O-mannosyltransferases, such as the distantly related transmembrane and tetratricopeptide repeat-containing proteins 1–4 (TMTC 1–4) and TMEM260 (Larsen et al, 2017a, 2017b, 2023). PMTs are part of the GT-C superfamily of ER-resident mannosyltransferases, which also include enzymes responsible for transferring mannoses for the synthesis of N-glycan precursors and glycosylphosphatidylinositol (GPI) anchors. All ER-resident mannosyltransferases use dolichol phosphate-mannose as a donor and share common catalytic motifs, as well as a common multi-transmembrane domain (TMD) architecture (Liu and Mushegian, 2003; Albuquerque-Wendt et al, 2019; Bloch et al, 2020; Alexander and Locher, 2023). PMTs have been linked to ERQC and are thought to possess protein chaperone functions and mannose-peptide binding capacity through their MIR (PMT, inositol 1,4,5-trisphosphate receptor [IP3R], and ryanodine receptor [RyR]) domains (Fukuda et al, 2001; Bai et al, 2019; Chiapparino et al, 2020). The roles of protein O-mannosylation in ERQC appear manifold and are related to protein folding and ER retention. Drug-induced inhibition of protein N-glycosylation has been shown to enhance global protein O-mannosylation as a compensatory mechanism, which, in turn, has been proposed to mitigate protein aggregation (Harty et al, 2001; Nakatsukasa et al, 2004). In non-stressed yeast cells, Pmt1 and Pmt2 promote ER retention of misfolded proteins and physically interact with central ERAD components (Goder and

¹Department of Genetics, University of Seville, Seville, Spain. ²Department of Molecular Genetics, Weizmann Institute of Sciences, Rehovot, Israel. ✉E-mail: llemus@us.es; vgoder@us.es

Melero, 2011). Finally, protein O-mannosylation has been shown to block (re)binding to ER chaperones in vitro and is proposed to be a mechanism to terminate energy-consuming folding cycles in the ER in vivo (Xu et al, 2013).

The misfolded yeast protein Gas1* has been used as a model to study ERAD of GPI-anchored proteins (GPI-APs) (Fujita et al, 2006). Gas1* is a mutant form of Gas1, an abundant protein in the secretory pathway. Unlike Gas1, which is moderately O-mannosylated by Pmt4, Gas1* is highly O-mannosylated, primarily by the Pmt1/2 complex (Gentzsch and Tanner, 1996; Hirayama et al, 2008; Goder and Melero, 2011; Sikorska et al, 2016). The degradation of Gas1* is complex and occurs dynamically through distinct cellular pathways (Lemus et al, 2023). A fraction is retained within the ER and subsequently routed to ERAD, while the remaining pool is rapidly exported from the ER and targeted to the vacuole for degradation (Fujita et al, 2006; Hirayama et al, 2008; Goder and Melero, 2011; Sikorska et al, 2016; Lemus et al, 2021). The cellular factors directing Gas1* to ERAD remain unknown.

We performed complementary genome-wide screens in the yeast model Saccharomyces cerevisiae to identify cellular components involved in the degradation of Gas1*. Our findings revealed an unexpected role for the evolutionarily conserved ER-resident proteins Pbn1 and Gpi14 (PIG-X and PIG-M in mammals) in both protein O-mannosylation and the ERAD pathway. Both proteins were previously known solely for their function as a mannosyltransferase in GPI anchor precursor synthesis. Our functional analyses, supported by phylogenetic data, suggest that Pbn1 and Gpi14 form a bifunctional heteromeric complex. In addition to their established role in GPI anchor biosynthesis, this complex contributes to the O-mannosylation of misfolded proteins containing a serine-rich region (SRR), thereby promoting their ERAD. Moreover, we found that Pbn1–Gpi14 supports the ERAD of a misfolded model protein lacking N-glycans through O-mannosylation. Collectively, our findings uncover protein O-mannosylation as an alternative glycan-based mechanism that is involved in targeting misfolded proteins for ERAD.

# Results

## Genome-wide yeast screens identify Pbn1 as a regulator for the degradation of misfolded GPI-AP Gas1*

We employed complementary genome-wide yeast screens to identify factors involved in ERAD of Gas1*. For a growth-based screen, we generated the chimeric protein Leu2-TMD-Gas1*, composed of Gas1* fused to Leu2 via the transmembrane domain (TMD) of the ER-resident protein Ted1 (Fig. 1A). In cells lacking genomic LEU2, involved in leucine biosynthesis, growth in the absence of this amino acid is predicted to occur only if the reporter fails to be degraded, such as upon genetic inhibition of ERAD (Fig. 1B). Consistent with this prediction, the ERAD mutant Δubc7, but not wild-type cells, supported growth in the absence of leucine (Fig. EV1A). Leu2-TMD-Gas1* was transformed into a collection of strains containing viable individual deletions or decreased abundance by mRNA perturbation (DAmP) alleles of essential genes, resulting in over 5800 transformants covering approximately 93% of all genes (Fig. 1B). Transformants were replica-plated into

media containing or lacking leucine, leading to the identification of 213 genes that affected reporter degradation, 36 of which encoded annotated ER proteins (Figs. 1B,E and EV1B and Dataset EV1). These included most Hrd1 complex components, an ERAD machinery required for Gas1* degradation, validating the screen's functionality (Fig. 1E).

For a complementary screen, we utilized GFP-Gas1* (Fig. 1C and Sikorska et al, 2016) aiming at identifying cellular factors whose overexpression leads to an increase in GFP-Gas1* degradation, thereby reducing cellular GFP fluorescence signal. GFP-Gas1* was first integrated into Δted1 cells to both reduce ER export and increase its routing to ERAD (Fig. EV1C and Sikorska et al, 2016). Next, the resulting strain was crossed with a whole-genome ORF expression library where each strain expresses a single, unique ORF with N-terminally fused mCherry under the control of the constitutively strong TEF2 promoter. This process yielded more than 5900 double-fluorescent strains, covering approximately 95% of all genes (Fig. 1D). Automated image acquisition and fluorescence signal quantification identified 153 genes that decreased the reporter signal by more than 20% upon strong expression, 10 of which were predicted or verified as ER protein-encoding genes (Fig. 1D,E and Dataset EV1). Using the aforementioned selection criteria, the essential gene PBN1 emerged as the sole overlapping hit in both screens (Fig. 1E).

## Pbn1 shares structural homology with PIG-X and affects the turnover of misfolded Gas1 variants with and without a GPI anchor

Pbn1 was initially reported to be an ER chaperone with roles in protein processing and quality control, though its mechanisms of action remained unknown (Naik and Jones, 1998; Subramanian et al, 2006). Complementation cloning in mammalian cells identified Pbn1 as the functional homolog of mammalian PIG-X, involved in the transfer of mannose from dolichol phosphate-mannose to GlcN-(acyl)PI in conjunction with its catalytic subunit PIG-M (Gpi14 in yeast) during GPI anchor precursor synthesis, linking Pbn1 to ER-resident mannosyltransferases (Ashida et al, 2005). When we superimposed structure predictions of Pbn1 and its human homolog PIG-X, we found a common C-terminal TMD, an evolutionarily conserved luminal domain proximal to the ER membrane, and a distal N-terminal domain present only in Pbn1, showing structural similarities between Pbn1 and PIG-X (Fig. 2A). These data support that Pbn1 is a component of the GPI-mannosyltransferase I (GPI-MT I) complex in yeast, likely underlying its essentiality. Using GFP-tagged Pbn1 for live-cell microscopy, we found that it co-localized with ER markers, whether expressed from its endogenous promoter (Fig. 2B) or upon overexpression from the TEF2 promoter used in our screen (Fig. EV2A). Based on these findings and our screening data, our initial hypothesis was that Pbn1 affects Gas1* degradation through its role in GPI metabolism (Wang et al, 2020).

Next, we directly measured protein degradation in vivo by blocking synthesis with cycloheximide (CHX) followed by western blot analysis of HA-tagged Gas1*. We used cells expressing the hypomorphic PBN1-DAmP allele identified in our screen alongside control cells. In general, the molecular weight (MW) of Gas1* at steady state was not uniform, and its degradation was preceded by an increase in its MW (Fig. 2C, lanes 1–4). By combining genetics

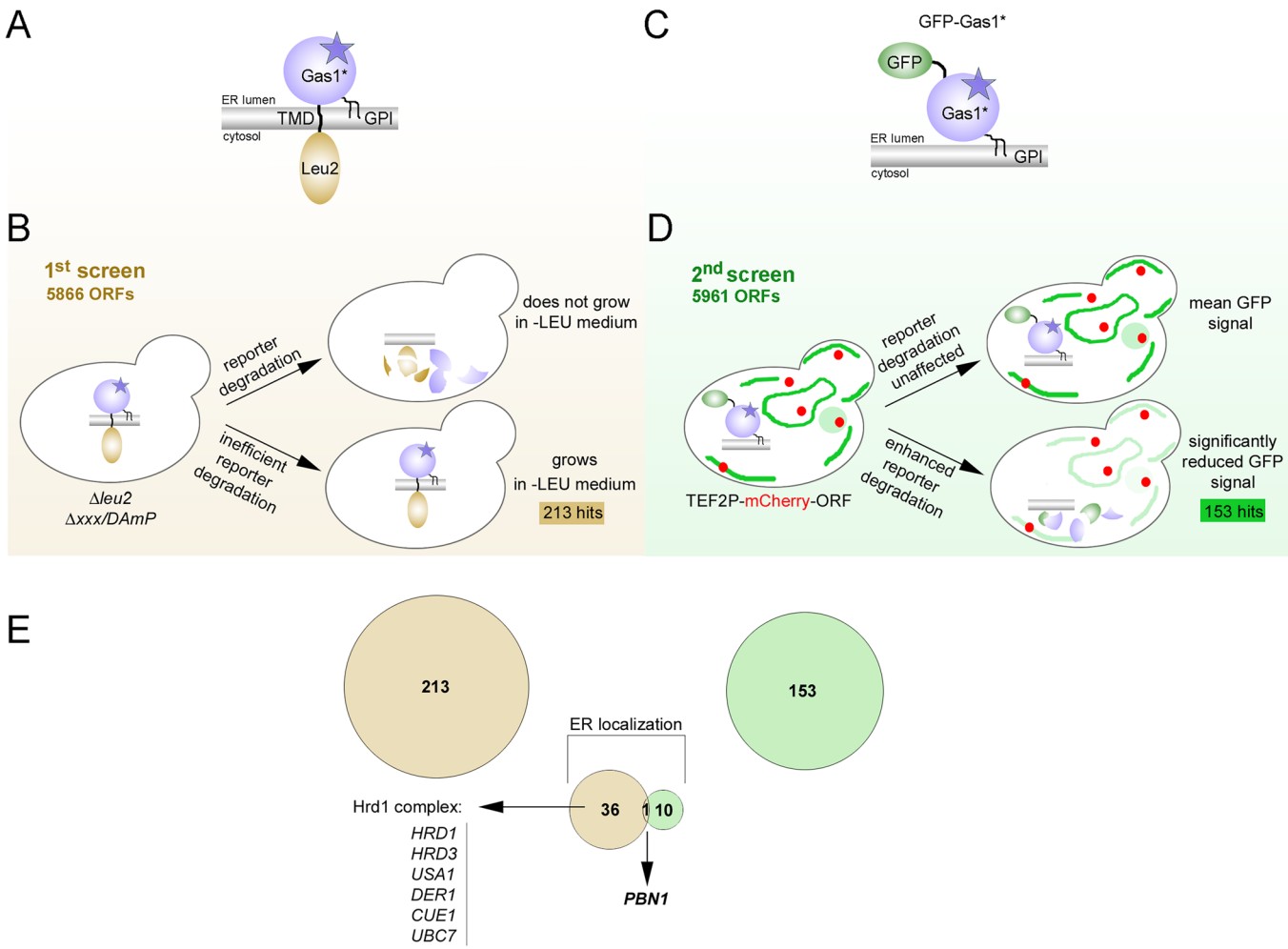

**Figure 1. Genome-wide yeast screens identify Pbn1 as a regulator for the degradation of misfolded GPI-AP Gas1\*.**

(A) Illustration of the Leu2-TMD-Gas1\* reporter and its topology in the ER membrane. Yeast Leu2 was fused to the N-terminus of Gas1\* via the transmembrane domain (TMD) of the ER-resident protein Ted1. Gas1\* contains a C-terminal GPI anchor (GPI). (B) Schematic illustration of the genome-wide screen using Leu2-TMD-Gas1\*. The collections of individual deletion mutants (Δxxx) and decreased abundance by mRNA perturbation (DAmP) mutants are auxotrophs for leucine (Δleu2). They were transformed with a URA3-bearing plasmid expressing the reporter from the weak GAL4 promoter. Transformants were grown in 96-well plates containing medium lacking uracil or both uracil and leucine. Efficient inhibition of Leu2-TMD-Gas1\* degradation allowed cells to grow in medium lacking leucine (hits). (C) Illustration of the GFP-Gas1\* reporter and its topology in the ER membrane. (D) A yeast query strain expressing genome-integrated GFP-Gas1\* from its endogenous promoter was crossed with the collection of mutants expressing N-terminal mCherry fusions to all individual ORFs under the control of the strong constitutive TEF2 promoter using synthetic genetic array technology. Double mutants were plated onto specific 96-well plates suited for fluorescence microscopy analysis, and individual wells were automatically imaged three times using high-throughput fluorescence microscopy. Automated thresholding identified mutants that showed significantly reduced relative GFP signal, indicative of enhanced reporter degradation (hits). (E) Summary of hits. Two hundred and thirteen genes were identified in the first screen, 36 of which are annotated for encoding ER proteins, including members of the Hrd1 complex (left). One hundred and fifty-three genes were identified in the second screen, ten of which are annotated for encoding ER proteins. PBN1 was the sole overlapping hit from both screens.

with treatment of cell lysates with the glycosidase PNGase F, we confirmed that most of it can be attributed to protein O-mannosylation by the Pmt1/2 complex, supporting previous data (Fig. EV2B; Fujita et al, 2006; Goder and Melero, 2011). The O-mannosylation-dependent increase in MW of Gas1\* occurs within the ER, rather than in the Golgi or other downstream organelles, as genetically blocking ER export did not produce changes in MW (Fig. EV2C,D). We also confirmed that deletion of the Pmt1/2 complex accelerated Gas1\* degradation (Fig. EV2E,F). It was previously shown that accelerated degradation under these conditions results from a loss of ER retention and an increase in the

fraction of Gas1\* that is directed to the vacuole (Hirayama et al, 2008; Goder and Melero, 2011; Sikorska et al, 2016).

In PBN1-DAmP cells, the overall degradation rate of Gas1\* was reduced, confirming our screen data (Fig. 2C,D). Notably, a subpopulation of Gas1\* showed no increase in MW (Fig. 2C, lanes 5–8, arrows). Further supporting the screen data, PBN1 over-expression enhanced Gas1\* degradation (Figs. 2E,F and EV2G). To test if the obtained phenotypes are linked to the presence of the GPI anchor on Gas1\*, we next used Gas1\*TMD as a negative control. This version of Gas1\* fails to attach to a GPI anchor and retains its C-terminal TMD (Sikorska et al, 2016). Although the degradation

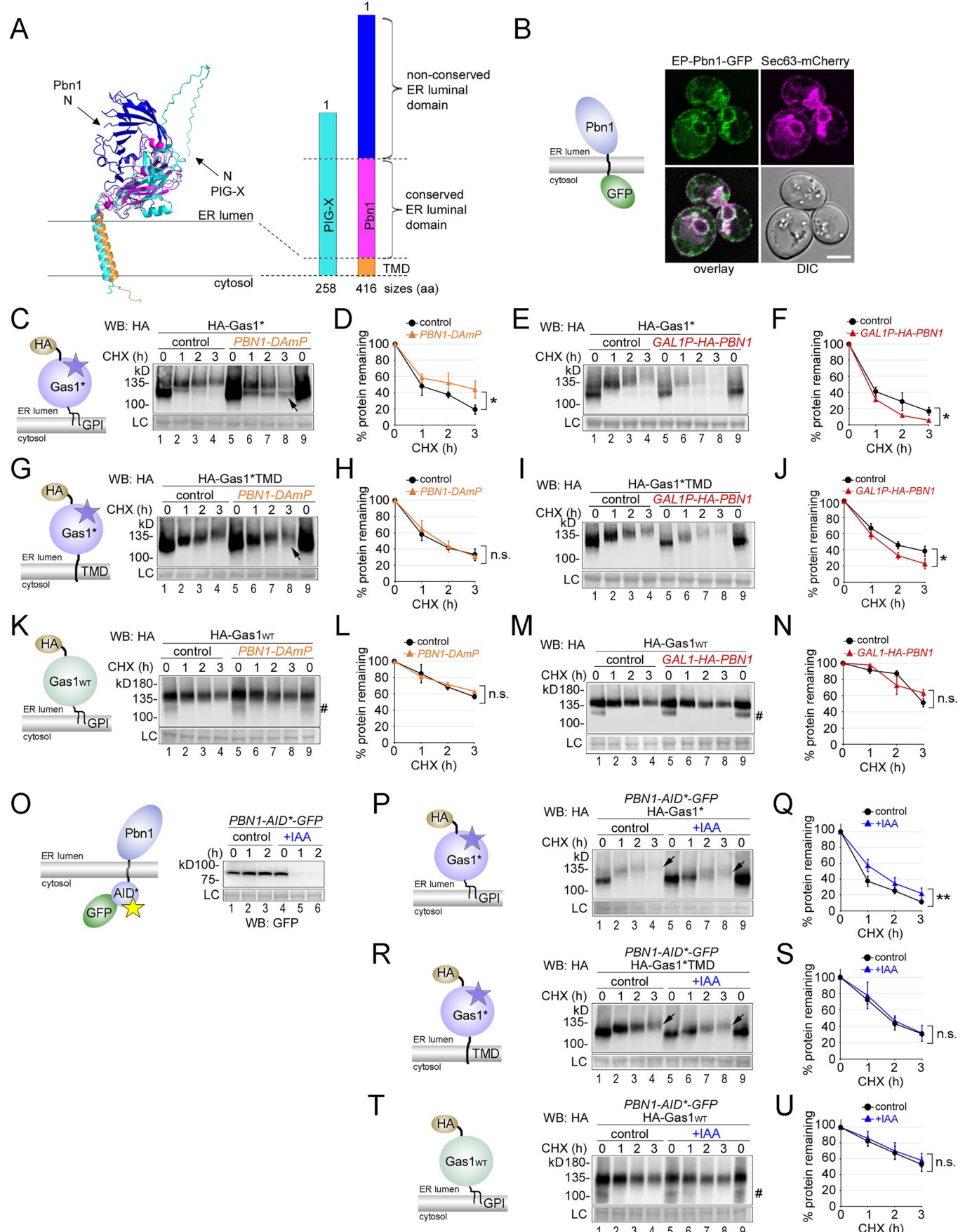

**Figure 2.  Pbn1 shares structural homology with PIG-X and affects the turnover of misfolded Gas1 variants with and without a GPI anchor.**

(A) AlphaFold structure predictions for yeast Pbn1 (blue/pink/orange) and human PIG-X (cyan) were superimposed using PyMOL (left panel). Protein lengths, their conserved and non-conserved domains, and their C-terminal TMDs are indicated with proportional bars (right panel). (B) Yeast cells co-expressing C-terminally GFP-tagged Pbn1 from its genomic locus and endogenous promoter (graphical depiction) and the ER marker Sec63-mCherry were analyzed by live-cell fluorescence microscopy and differential interference contrast (DIC) microscopy. Scale bar: 3 μm. (C) Control cells and cells containing the *PBN1-DAmP* allele were used to express plasmid-borne HA-tagged Gas1* (as depicted) for cycloheximide (CHX) shut-off experiments. Cells were lysed at the indicated time points after application of CHX, and remaining HA-Gas1* was measured by SDS-PAGE and western blotting (WB) with antibodies against HA. Membrane staining with Ponceau served as a loading control (LC). The arrow indicates a subpopulation of HA-Gas1* with lower MW in *PBN1-DAmP* cells compared to control cells. (D) Quantifications of results from experiments shown in (C). Total remaining protein (upper and lower bands combined) has been quantified. (E, F) Control cells and cells that expressed genomically HA-tagged Pbn1 from the *GAL1* promoter (*GAL1P-HA-PBN1*) were used to express plasmid-borne HA-Gas1*. Experimental procedures and quantifications are described in (C, D) with the exception that cells were grown in a medium supplemented with galactose as carbon source for the expression of HA-Pbn1. (G–J). As in (C–F), but expressing plasmid-borne HA-Gas1*TMD. HA-Gas1*TMD contains the N428Q mutation, which prevents TMD removal and GPI anchor attachment, thereby rendering HA-Gas1*TMD a TM protein. The arrow indicates a subpopulation of HA-Gas1*TMD with lower MW in *PBN1-DAmP* cells compared to control cells. (K–N) As in (C–F), but expressing plasmid-borne HA-tagged wild-type Gas1 (HA-Gas1$_{WT}$). Species with lower MW at times zero (#) represent the non-glycosylated ER form of HA-Gas1$_{WT}$ (Sikorska et al, 2016). (O) Cells expressing genomic Pbn1 C-terminally tagged with the degron AID* in combination with GFP (graphical depiction) were supplemented with auxin (indole-3-acetic acid (IAA)) or only with solvent (control). Equal amounts of cells were removed at the indicated time points after supplementation and analyzed by SDS-PAGE and WB with antibodies against GFP. (P, Q) HA-tagged Gas1* was expressed in cells containing genomic *PBN1-AID*-GFP*. Three hours prior to the start of the CHX shut-off, cells were supplemented with IAA or only with solvent (control). Cells were processed as described in (C, D). Arrows (lanes 4 and 8) serve to indicate the differences in MW of the protein three hours after addition of CHX, relative to time zero (lanes 5 and 9). (R, S) As in (P, Q), but expressing plasmid-borne HA-tagged Gas1*TMD. Arrows (lanes 4 and 8) serve to indicate the differences in MW of the protein three hours after addition of CHX, relative to time zero (lanes 5 and 9). (T, U) As in (P, Q), but expressing plasmid-borne HA-tagged wild-type Gas1 (HA-Gas1$_{WT}$). Species with lower MW at times zero (#) represent the non-glycosylated ER form of HA-Gas1$_{WT}$ (Sikorska et al, 2016). Data information: Number of experiments (*n*) signifies biological replicates. Error bars in graphs represent the standard deviation from the mean. Statistical significance for all experiments involving degradation rates was calculated using two-way ANOVA (alpha = 0.05) with Šidák's correction for multiple comparisons, obtaining the following *p* values. (D): (*n* = 4, *p* = 0.0254), (F): (*n* = 3, *p* = 0.0281), (H): (*n* = 4, *p* = 0.4590), (J): (*n* = 3, *p* = 0.0304), (L): (*n* = 2, *p* = 0.5062), (N): (*n* = 2, *p* = 0.4500), (Q): (*n* = 4, *p* = 0.0036), (S): (*n* = 3, *p* = 0.6341), (U): (*n* = 4, *p* = 0.9386). Difference in degradation rate: n.s. = not significant; *$p < 0.05$; **$p < 0.01$. Source data are available online for this figure.

rate of Gas1*TMD was unaffected in *PBN1-DAmP* cells (Fig. 2G,H), we observed significantly faster degradation of Gas1*TMD in cells overexpressing *PBN1* (Fig. 2I,J). In addition, we consistently detected a small subpopulation in *PBN1-DAmP* cells without an increase in MW which was absent in control cells (Fig. 2G, lane 8, arrow; Fig. 2H). It is unlikely that this is an indirect effect of a potentially altered GPI metabolism, since expression of Gas1* in Δ*ted1* and Δ*cwh43* deletion mutants, which severely disrupt GPI metabolism by blocking glycan and lipid remodeling of GPI anchors and produce a strong UPR response in case of Δ*ted1* cells (Jonikas et al, 2009), showed the previously reported stabilization of Gas1* but no O-mannosylation defects (Fig. EV2H and Sikorska et al, 2016). No changes in the MW or the turnover rate were observed for HA-tagged wild-type Gas1 in cells with either reduced or increased *PBN1* levels (Fig. 2K–N).

To assess whether the phenotypes observed in *PBN1-DAmP* cells are an indirect result of long-term cellular adaptations, we used experiments with acute depletion of Pbn1. To this end, we C-terminally tagged Pbn1 with the auxin-inducible degron AID* in combination with GFP at its genomic locus. Application of auxin led to rapid depletion of Pbn1-AID*GFP (Fig. 2O). Confirming the data obtained with *PBN1-DAmP* cells, depletion of Pbn1 resulted in reduced O-mannosylation of Gas1* and Gas1*TMD compared to control conditions (Fig. 2P,R lanes 4 and 8, arrows), whereas the MW of wild-type Gas1 remained unchanged (Fig. 2T). In addition, the degradation of Gas1* was significantly reduced upon Pbn1 depletion, whereas it remained unchanged for Gas1*TMD and wild-type Gas1 (Fig. 2Q,S,U).

Together, these findings indicate that differential expression of Pbn1 specifically regulates the O-mannosylation and degradation of misfolded Gas1 variants, even those lacking a GPI anchor, suggesting additional cellular functions for Pbn1 beyond its established role in GPI anchor biosynthesis. In an appealing working model, Pbn1 function might directly or indirectly enhance

O-mannosylation of misfolded proteins in order to promote their degradation. These results prompted further exploration of cellular Pbn1 functions.

## Pbn1 function is connected to both protein O-mannosylation and ERAD

In a first approach to address the role of Pbn1 in protein O-mannosylation for degradation, we generated several versions of the N-glycosylated misfolded ER model protein CPY*. CPY* is a luminal protein and typically degraded independently of O-mannosylation. Consistent with this, it displayed a uniform MW and its degradation rate was unaffected in both *PBN1-DAmP* cells and cells overexpressing *PBN1* (Fig. 3A–D).

By simply fusing the C-terminal domain of Gas1*TMD, which includes its TMD and part of its proximal SRR, to the C-terminus of CPY*, we generated the diagnostic construct CPY*SRR_TMD (Fig. 3E, SRR in yellow). The SRR of Gas1* is a predicted target for O-mannosylation and is thought to account for the majority of the observed increase in MW (Nuoffer et al, 1993; Fujita et al, 2006). Consistent with CPY*SRR_TMD becoming O-mannosylated, the MW of CPY*SRR_TMD at steady state was not uniform and its degradation was preceded by an increase in MW (Fig. 3F, lanes 1–4). Because the SRR-TMD domain appended to CPY* is devoid of N-glycosylation sites, the observed changes in MW do not originate from N-glycosylation. Notably, deletion of the Pmt1/2 complex reduced the MW increase of CPY*SRR_TMD, confirming that the protein was O-mannosylated (Fig. EV3A). Additionally, the deletion of the Pmt1/2 complex increased the degradation rate of CPY*SRR_TMD, mirroring the effects seen with Gas1* (Figs. EV3A,B and EV2E,F).

In *PBN1-DAmP* cells, a fraction of CPY*SRR_TMD showed no increase in MW, suggesting inefficient O-mannosylation (Fig. 3F, lanes 5–9, arrow). At the same time, the overall degradation rate of

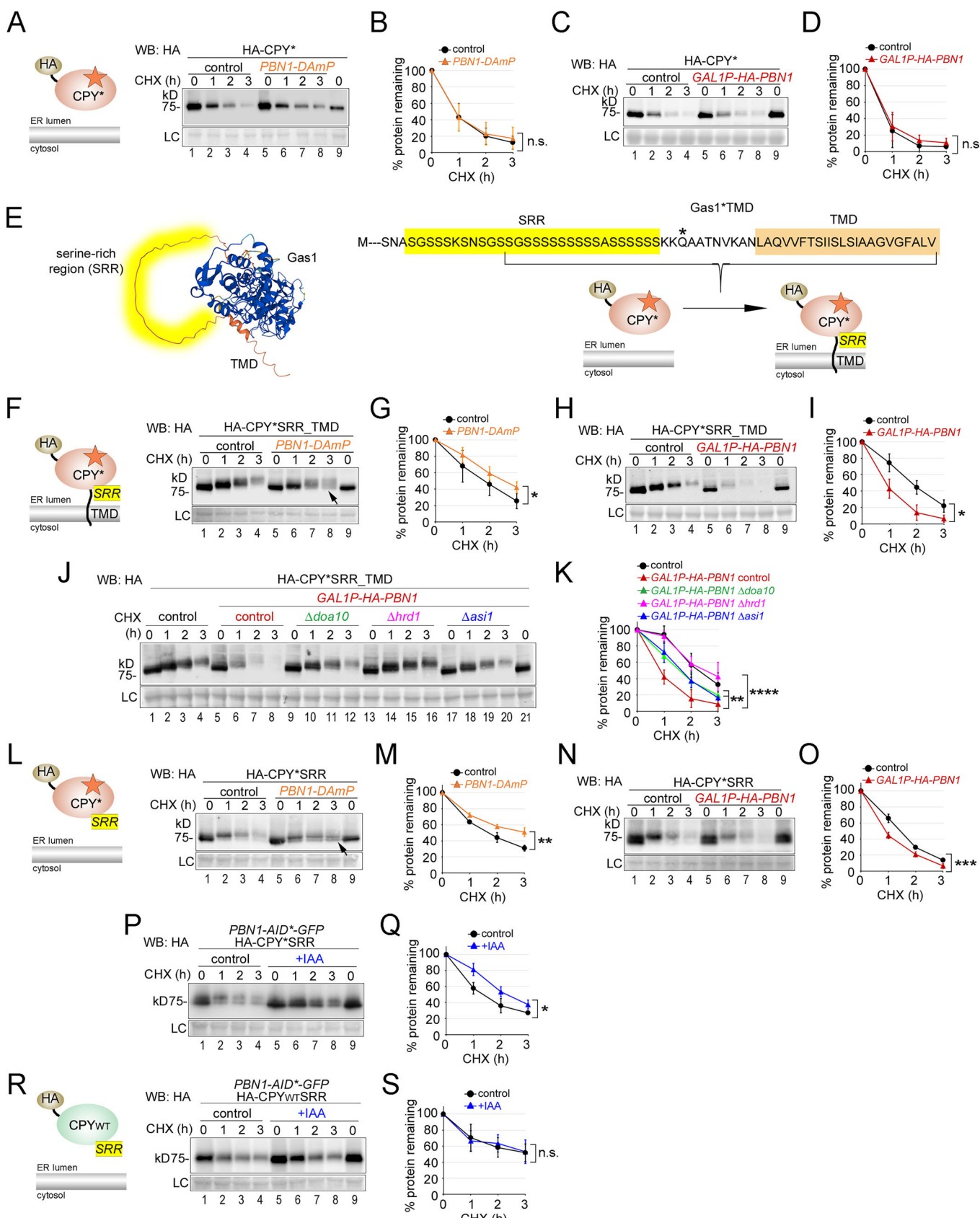

◄

**Figure 3. Pbn1 function is connected to both protein O-mannosylation and ERAD.**

(A) Control cells and cells containing the *PBN1-DAmP* allele were used to express plasmid-borne HA-tagged CPY* (depiction) for CHX shut-off experiments. (B) Quantifications of results from experiments shown in (A). (C, D) Control cells and cells containing *GAL1P-HA-PBN1* were used to express plasmid-borne HA-CPY* for CHX shut-off experiments in medium supplemented with galactose as a carbon source for the expression of HA-Pbn1. (E) Left panel: AlphaFold structure prediction of wild-type Gas1 with the C-terminal unstructured serine-rich region (SRR) (yellow) upstream of the TMD (orange). Right panel: The C-terminal region of Gas1*TMD is shown in single-letter amino acid code with the SRR (Ser497-Ser525) and the TMD. The N428Q mutation that prevents GPI anchor attachment is marked with an asterisk (*). To generate HA-CPY*SRR_TMD, the indicated sequence of Gas1*TMD (in brackets) was fused with the C-terminus of HA-CPY*. (F–I) As in (A–D), but with plasmid-borne HA-CPY*SRR_TMD used for expression in the indicated control and mutant strains. The arrow marks a band with a lower MW that is absent in control cells. (J, K) Control cells and mutants lacking the indicated ERAD components and containing *GAL1P-HA-PBN1* where indicated were used to express plasmid-borne HA-CPY*SRR_TMD. Cells were grown overnight in medium containing raffinose and then diluted into medium containing galactose for the expression of HA-Pbn1 three hours prior to the application of CHX. (L–O) As in (A–D), but with plasmid-borne HA-CPY*SRR used for expression in the indicated control and mutant strains. The arrow indicates an increased protein population with a lower MW compared to control cells. (P, Q) HA-CPY*SRR was expressed in cells containing genomic *PBN1-AID*-GFP*. Three hours prior to the start of the CHX shut-off, cells were supplemented with IAA or only with solvent (control). (R, S) Like in (P, Q) but expressing HA-tagged wild-type CPY containing the C-terminal SRR (HA-CPYwtSRR). Data information: Number of experiments (n) signifies biological replicates. Error bars in graphs represent the standard deviation from the mean. Statistical significance for all experiments involving degradation rates was calculated using two-way ANOVA (alpha = 0.05) with Šidák's correction for multiple comparisons, obtaining the following p values. (B): (n = 4, p = 0.9478), (D): (n = 3, p = 0.9466), (G): (n = 11, p = 0.0247), (I): (n = 3, p = 0.0109), (K): (Δdoa10: n = 3, p = 0.0013; Δhrd1: n = 3, p = 0.00009; Δasi1: n = 3, p = 0.0011), (M): (n = 3, p = 0.0017), (O): (n = 3, p = 0.0004), (Q): (n = 3, p = 0.024), (S): (n = 3, p = 0.9386). Difference in degradation rate: n.s. = not significant; *p < 0.05; **p < 0.01; ***p < 0.001; ****p < 0.0001. Source data are available online for this figure.

the protein was reduced (Fig. 3F,G). Conversely, the degradation rate of CPY*SRR_TMD was accelerated with *PBN1* overexpression (Fig. 3H,I). To determine whether the increased degradation rate of CPY*SRR_TMD with Pbn1 overexpression is linked to ERAD, we repeated the experiments in specific mutant backgrounds. Yeast contains three distinct ERAD complexes: the Hrd1-, the Doa10-, and the Asi-complex; for review, see (Krshnan et al, 2022). Deletion of any of the ERAD complexes reduced the degradation rate, with the strongest reduction observed in *Δhrd1* cells (Fig. 3J,K). In *Δemp24* cells, which are proficient in ERAD but defective in vesicular ER export of a subset of cellular cargo, thereby affecting ER homeostasis and inducing the UPR (Belden and Barlowe, 2001), CPY*SRR_TMD degradation was not significantly different from that in control cells (Fig. EV3C,D). These results rule out the possibility that protein stabilization observed in ERAD mutants is an indirect consequence of altered ER homeostasis and/or UPR activation. Instead, they support a direct role for Pbn1 in the degradation of misfolded proteins via the ERAD pathway.

To distinguish potential effects of the TMD on the degradation properties of CPY*SRR_TMD, we next generated CPY*SRR, which lacks the TMD. We observed phenotypes comparable to those seen with CPY*SRR_TMD when expressed in cells with either reduced or elevated *PBN1* levels (Fig. 3L–O). We also detected a decreased degradation rate of CPY*SRR upon acute depletion of Pbn1 (Fig. 3P,Q). Importantly, the slower degradation of a control construct with the SRR fused to wild-type CPY was not affected upon Pbn1 depletion and showed no significant increase in MW, suggesting no or negligible O-mannosylation (Fig. 3R,S). Taken together, these phenotypes support a functional link between Pbn1 and the O-mannosylation of an SRR on misfolded proteins.

Interestingly, the degradation kinetics of SRR-containing misfolded proteins like Gas1* or CPY*TMD_SRR showed opposite tendencies in cells depleted of either the Pmt1/2 complex or with reduced Pbn1 expression levels (Fig. EV3A,B and Goder and Melero, 2011). Although protein O-mannosylation was decreased in all cases, we generally observed faster degradation in cells devoid of the Pmt1/2 complex, but slower degradation in cells expressing less Pbn1. We confirmed that this disparity resulted from differences in the quantity of protein that exited the ER and was subsequently routed to the vacuole in each case. Whereas the

absence of the Pmt1/2 complex caused increased routing of Gas1* and CPY*SRR_TMD to the vacuole for rapid degradation, no such effect was observed in *PBN1-DAmP* cells (Fig. EV3E,F and Goder and Melero, 2011). These findings support earlier observations that PMTs possess an ER retention function (Goder and Melero, 2011) and suggest that Pbn1 does not promote ER retention.

## Pbn1 co-localizes with Gpi14 in vivo

Complementation assays predicted Pbn1 to be functionally connected to the ER mannosyltransferase and GT-C superfamily member protein Gpi14 for its role in GPI anchor biosynthesis (Ashida et al, 2005). To investigate whether Pbn1 is associated with Gpi14 in vivo, we utilized the observation that N-terminal tagging of Pbn1 with GFP caused its accumulation in ER puncta, independent of its expression levels (Fig. EV4A,B). Whereas Gpi14-tdimer showed uniform expression across the perinuclear and peripheral ER, its co-expression with GFP-Pbn1 led to the nearly complete co-localization of the visible cellular pool of Gpi14 with Pbn1 in ER puncta (Fig. 4A,B). The co-expression of the tagged versions of these essential proteins did not compromise their main functions, as no growth defects were observed under these conditions, even in the presence of the N-glycosylation inhibitor and ER stress inducer tunicamycin (Fig. EV4C). These data indicate that, beyond their previously reported functional connection, Pbn1 co-localizes with Gpi14 in vivo. The nature of this co-localization supports the notion that both components are part of a shared complex.

The AlphaFold multimer structure of the simplest stoichiometric complex between Pbn1 and Gpi14, a heterodimer, suggests how Pbn1 is functionally connected to Gpi14 (Fig. 4C). Pbn1 provides a large ER luminal domain to Gpi14 and forms an interface with the conserved catalytic DxD motif of Gpi14 for the transfer of mannoses (Fig. 4C, motif depicted in green, and Wiggins and Munro, 1998). Among the four ER mannosyltransferases involved in GPI anchor biosynthesis in yeast, Pbn1-Gpi14 has a unique molecular architecture and provides an unusually large ER luminal domain in the form of Pbn1 to the catalytic Gpi14, whereas the other enzymes have significantly smaller luminal domains or lack them entirely (Fig. EV4D). In fact, the Pbn1-Gpi14

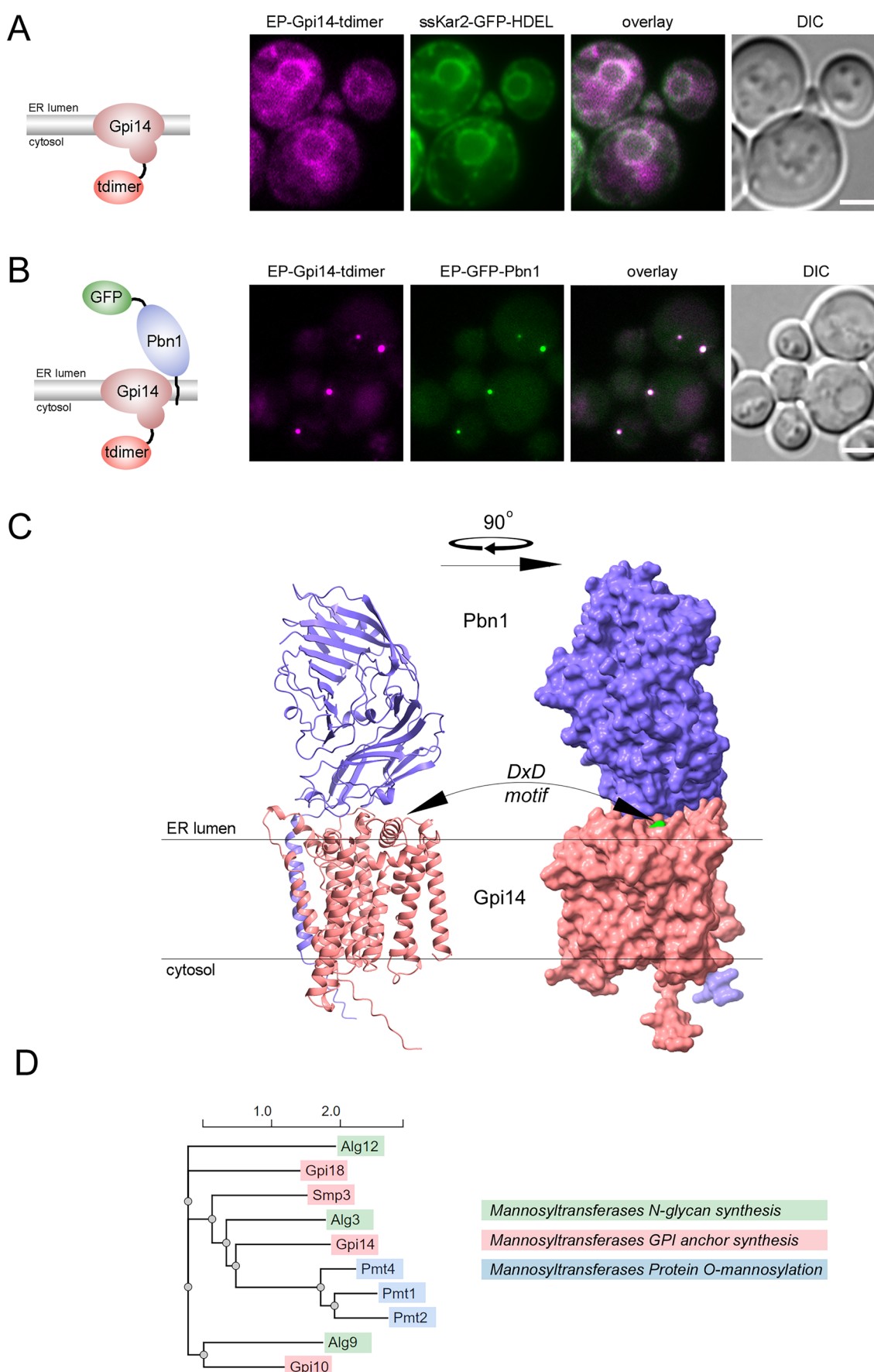

**Figure 4. Pbn1 co-localizes with Gpi14 in vivo.**

(A) Yeast cells co-expressing C-terminally tdimer-tagged genomic Gpi14 (as depicted) and the ER marker ssKar2-GFP-HDEL were analyzed by live-cell fluorescence microscopy and DIC microscopy. Scale bar: 2 µm. (B) Yeast cells co-expressing C-terminally tdimer-tagged genomic Gpi14 and N-terminally GFP-tagged genomic Pbn1 (depiction) were analyzed by live-cell confocal fluorescence microscopy and DIC microscopy. Scale bar: 3 µm. (C) AlphaFold multimer prediction for a heterodimer containing Pbn1 (purple) and Gpi14 (pink). The catalytic DxD motif (green) in Gpi14 is localized within a cleft near the luminal side of the ER membrane oriented toward the conserved luminal domain of Pbn1. (D) Phylogram from a phylogenetic analysis of the amino acid sequences of the indicated members of the ER-resident GT-C superfamily of O-mannosyltransferases, using NGPhylogeny.fr (Lemoine et al, 2019). The color bars refer to annotated enzyme functions. Source data are available online for this figure.

architecture is more similar to PMTs and distantly related to TMTCs, which contain large and unrelated MIR and tetratricopeptide repeat-containing domains, respectively, for protein and peptide binding (Larsen et al, 2017a; Chiapparino et al, 2020).

Next, we conducted a phylogenetic analysis to assess the evolutionary distance of Gpi14 to the other major ER mannosyltransferases of the GT-C superfamily. Notably, based on the amino acid sequences of all shown family members, Gpi14 is positioned closest to the PMTs (Fig. 4D). Together, these data support a connection between protein O-mannosylation and Pbn1-Gpi14.

## Pbn1-Gpi14 promotes O-mannosylation and ERAD of a misfolded non-N-glycosylated model protein

The data that suggested a role for Pbn1 in protein O-mannosylation and ERAD originated from our analyses of misfolded model proteins containing an SRR. However, SRRs are only found in a small subset of cellular proteins and are mostly studied in bacteria (Cinar et al, 2024). Pbn1 function could thus be confined to a few specific substrates. Conversely, protein O-mannosylation is not restricted to SRRs and can occur on individual serine or threonine residues throughout a polypeptide, a feature known to play a role under certain ER stress conditions, such as those caused by chemical inhibition of N-glycosylation (Harty et al, 2001). The presence of N-glycans may therefore mask or reduce the necessity for O-mannosylation-dependent ERAD for most misfolded ER proteins.

We therefore extended our analysis to the non-N-glycosylated CPY* variant ΔNg-CPY*, which lacks all its four N-glycosylation sites and does not contain an SRR (Kostova and Wolf, 2005). Like nearly all proteins, it retains numerous serines and threonines scattered throughout its sequence (Fig. 5A). Unlike CPY*, ΔNg-CPY* was tunicamycin-insensitive, confirming its lack of N-glycosylation (Fig. 5B,C). ΔNg-CPY* was degraded less efficiently than CPY* (Fig. 5D, lanes 1–4, Fig. 5E, compare with Fig. 3, lanes 1–4, Fig. 3B) and stabilized in Δhrd1 cells, indicating that it was degraded via Hrd1-dependent ERAD (Fig. 5D, lanes 5–9, Fig. 5E). This demonstrated that ΔNg-CPY* was not aggregated and remained retrotranslocation-competent. Unlike CPY*, ΔNg-CPY* displayed a subpopulation with higher MW (Fig. 5D, lanes 1–4, arrows). After treatment of the cell lysate with jack-bean mannosidase, the subpopulation with higher MW was no longer detectable, showing that ΔNg-CPY* undergoes O-mannosylation (Fig. 5F). In support, we could isolate the ΔNg-CPY* subpopulation with higher MW using bead-coupled concanavalin A (conA), which only binds glycosylated proteins (Fig. 5G, lanes 1 and 2). ΔNg-CPY* with lower MW and the non-glycosylated control protein CFTR both failed to bind to conA (Figs. 5G and EV5A). In

Δhrd1 cells, the precipitable mannosylated forms of ΔNg-CPY* were heterogenous and included higher MW species, indicating that O-mannosylation is dynamic and precedes retrotranslocation, which is impaired in Δhrd1 cells (Fig. 5G, compare lanes 2 and 4).

To assess the functional connection of Pbn1 and Gpi14 in ΔNg-CPY* O-mannosylation and ERAD, we performed CHX shut-off experiments upon auxin-mediated depletion of AID*-tagged genomic versions of Pbn1 and Gpi14. Depletion of either component resulted in ΔNg-CPY* stabilization, primarily through the accumulation of the non-mannosylated form of the protein (Fig. 5H–K). As a control, the degradation of CPY* remained unaffected upon acute depletion of Pbn1, indicating that ERAD capacity was not generally compromised under these conditions (Fig. 5L,M). Equivalent results were obtained with cells containing either the PBN1-DAmP or GPI14-DAmP allele (Fig. EV5B,C). Together, these data support that Pbn1 and Gpi14 are functionally connected in protein O-mannosylation and ERAD of a misfolded ER protein that lacks both N-glycans and an SRR. Interestingly, we did not detect a contribution from canonical PMTs to the O-mannosylation and ERAD of ΔNg-CPY* (Fig. EV5D,E). Cells depleted of Pbn1 or Gpi14 exhibited increased sensitivity to tunicamycin, consistent with a role in maintaining ER homeostasis under N-glycosylation stress (Fig. EV5F).

## Functional analysis of Pbn1

Next, we aimed to generate Pbn1 mutants exhibiting phenotypes associated with protein O-mannosylation and ERAD of ΔNg-CPY*. Structural predictions indicated that the conserved membrane-proximal domain adopts a six-strand immunoglobulin (Ig)-like fold in both yeast Pbn1 and human PIG-X (Fig. 6A). In contrast, the distal region exhibits species-specific differences. In yeast, it forms a rigid, twisted, antiparallel β-sheet fold, whereas in PIG-X, it is significantly shorter and unstructured (Fig. 6A). This lack of structural conservation suggests evolutionary transitions and may relate to the bifunctionality of Pbn1-Gpi14.

A database search across kingdoms for folds resembling the distal Pbn1 region identified a similar domain exclusively in yeast Gpi16 and its homologs, including human PIG-T. Interestingly, this protein appears related to Pbn1, as it also contains an IgG-like fold, albeit with eight strands (Fig. EV6A). Despite these structural commonalities, Pbn1 and Gpi16 share only 18% sequence identity, indicating a distant common ancestor (Fig. EV6B). Gpi16/PIG-T is a component of the pentameric GPI transamidase complex, where the twisted antiparallel β-sheet provides extensive contact regions with three other subunits, suggesting that the domain acts as a scaffold for protein-protein interactions (Xu et al, 2023).

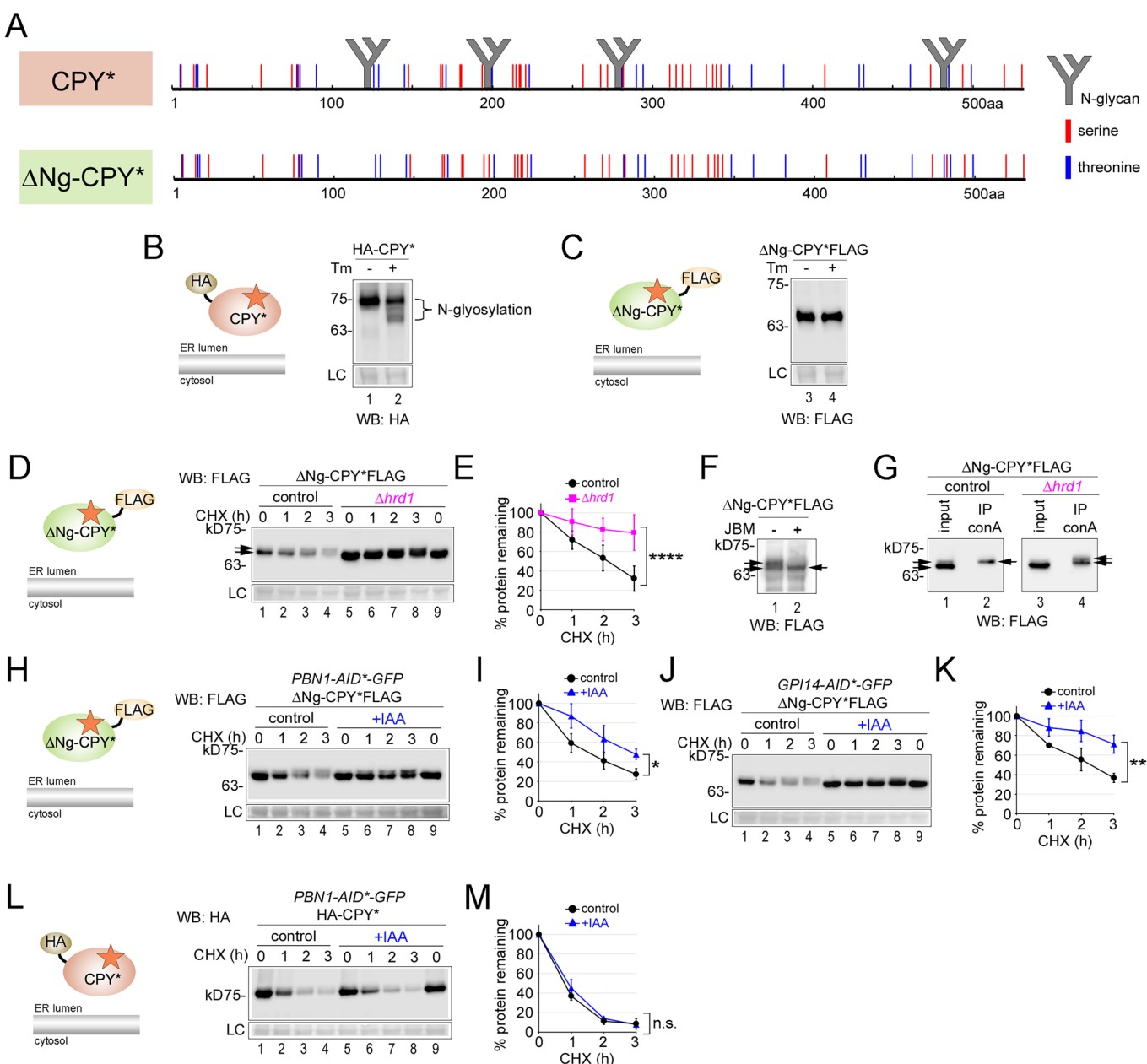

**Figure 5. Pbn1-Gpi14 promotes O-mannosylation and ERAD of a misfolded non-N-glycosylated model protein.**

(A) Schematic illustrating the properties of CPY* and ΔNg-CPY* protein sequences. CPY* is modified with four N-linked glycans at the indicated positions, whereas ΔNg-CPY* contains mutated N-glycosylation sites and lacks N-glycans. Both proteins contain identical serines and threonines distributed throughout their sequences, as indicated, but lack an SRR. (B) Lysates from cells expressing HA-tagged CPY* (graphical depiction) were analyzed by western blot (WB) analysis before and after cells were grown in the presence of 1 μg/ml tunicamycin (Tm) for one hour. (C) As in (B) using cells expressing FLAG-tagged ΔNg-CPY* (graphical depiction). (D) Control and Δhrd1 cells were used to express plasmid-borne ΔNg-CPY*FLAG for CHX shut-off experiments. The arrows indicate the presence of at least two protein populations with different MWs. (E) Quantifications of results from experiments shown in (D). (F) Cells expressing ΔNg-CPY*FLAG were lysed, and the lysate was incubated either with jack-bean-mannosidase (+) or with solvent (−) for 4 h, followed by WB analysis. The arrows indicate that the visible band with higher MW was sensitive to mannosidase treatment. (G) Control and Δhrd1 cells expressing ΔNg-CPY*FLAG were lysed, and the lysate was incubated for 3 h with concanavalin A (conA)-sepharose, followed by precipitation and WB analysis. Arrows in lanes 1 and 3 indicate the different molecular weight forms in the input fraction (starting material), while arrows in lanes 2 and 4 show the higher molecular weight forms recovered after precipitation. (H) Cells expressing genomic PBN1-AID*-GFP were used to express plasmid-borne ΔNg-CPY*FLAG for CHX shut-off experiments. Three hours prior to the start of the CHX shut-off, cells were supplemented with IAA or only with solvent (control). (I) Quantification of results shown in (H). (J, K) As in (H, I) but with cells expressing genomic GPI14-AID*-GFP. (L, M) As in (H, I) but with cells expressing plasmid-borne HA-CPY*. Data information: Number of experiments (n) signifies biological replicates. Error bars in graphs represent the standard deviation from the mean. Statistical significance for all experiments involving degradation rates was calculated using two-way ANOVA (alpha = 0.05) with Šidák's correction for multiple comparisons, obtaining the following p values. (E): (n = 5, p = 0.00009), (I): (n = 4, p = 0.0114), (K): (n = 4, p = 0.0072), (M): (n = 2, p = 0.2753). Difference in degradation rate: n.s. = not significant; *p < 0.05; **p < 0.01; ****p < 0.0001. Source data are available online for this figure.

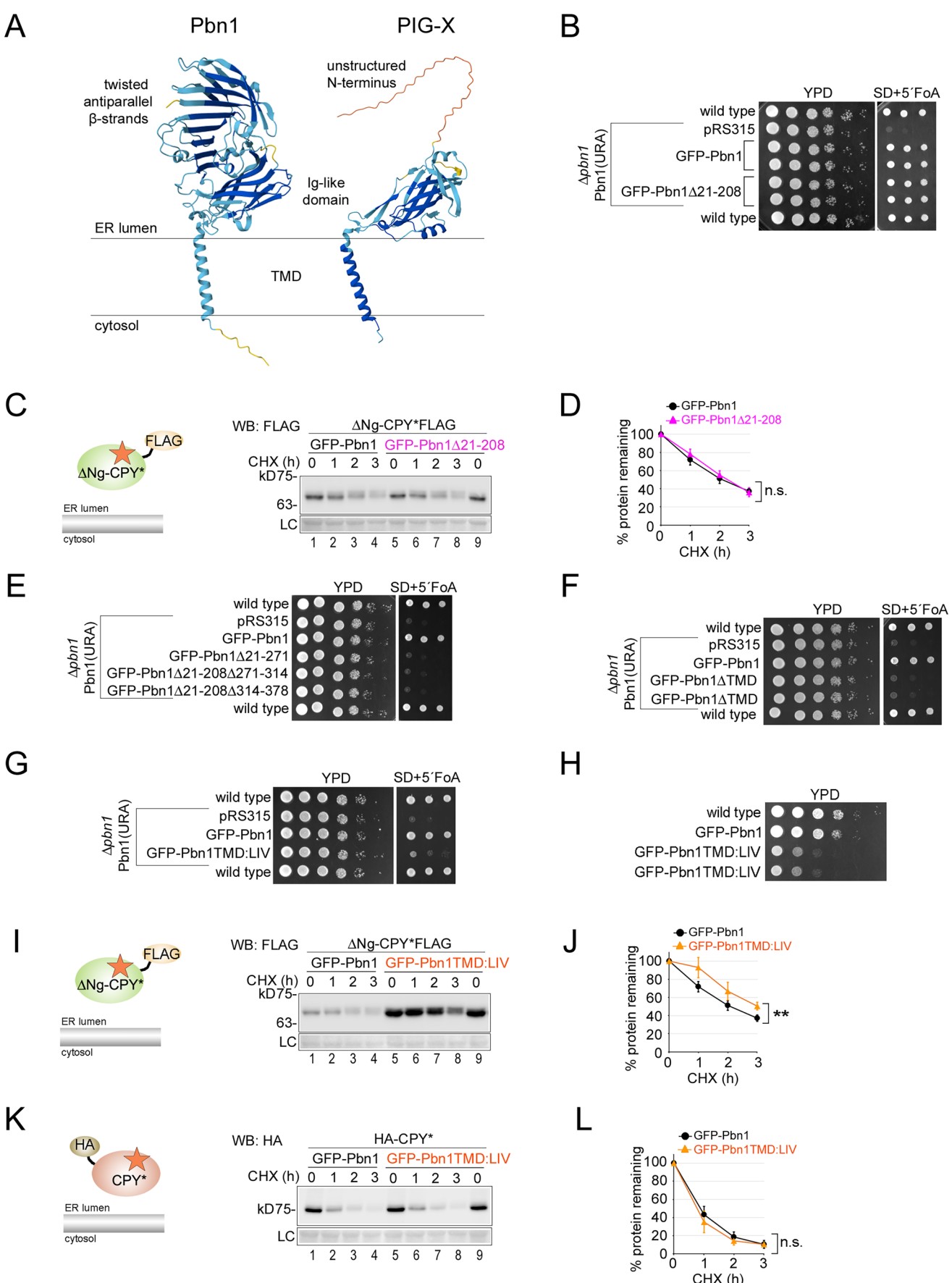

**Figure 6.  Functional analysis of Pbn1.**

(A) Alphafold3 folding predictions for Pbn1 and its human homolog PIG-X. The distal N-terminal regions of Pbn1 and PIG-X adopt distinct folds, whereas the conserved membrane-proximal regions in both proteins exhibit an immunoglobulin (Ig)-like fold. (B) Cellular growth assay. Pbn1 was expressed from its endogenous promoter in a *URA3*-containing plasmid in the Δ*pbn1* background. The indicated GFP-Pbn1 variants were expressed from a *LEU2*-containing plasmid, while the pRS315 plasmid served as a negative control without any insert. Wild-type cells were included as a positive control. For identically described constructs, cells transformed with independently generated plasmids were used. Cells were serially diluted to equal concentrations and spotted onto plates containing rich medium (YPD). Plates were incubated at 30 °C for 2 days before imaging. Similarly, the same dilutions were spotted onto plates containing 5-fluoroorotic acid (5-FOA) and incubated at 30 °C for 2 days. (C) CHX shut-off experiments using plasmid-borne GFP-Pbn1 and its mutant lacking the distal N-terminal region in the Δ*pbn1* background in cells expressing plasmid-borne ΔNg-CPY*FLAG. (D) Quantification of results from (C). (E–G) Experiments as those described in (B) were performed using the indicated mutant GFP-Pbn1 versions. (H) Serial dilution assay. Wild-type cells, as well as cells expressing plasmid-borne GFP-Pbn1 or GFP-Pbn1TMD:LIV in the Δ*pbn1* background, were spotted onto plates containing rich medium (YPD). Plates were incubated at 30 °C for 2 days prior to imaging. Two independently generated clones expressing GFP-Pbn1TMD:LIV were used. (I, J) As in (C, D) but with cells expressing GFP-Pbn1TMD:LIV and ΔNg-CPY*FLAG. (K, L) As in (C, D) but with cells expressing GFP-Pbn1TMD:LIV and HA-CPY*. Data information: Number of experiments (*n*) signifies biological replicates. Error bars in graphs represent the standard deviation from the mean. Statistical significance for all experiments involving degradation rates was calculated using two-way ANOVA (alpha = 0.05) with Šidák's correction for multiple comparisons, obtaining the following *p* values. (D): (*n* = 4, *p* = 0.3039), (J): (*n* = 4, *p* = 0.0033), (L): (*n* = 4, *p* = 0.3645). Difference in degradation rate: n.s. = not significant; **$p < 0.01$. Source data are available online for this figure.

We generated deletion mutants of plasmid-borne GFP-Pbn1 in the Δ*pbn1* background and first evaluated their effects on cell viability through 5-fluoroorotic acid (5-FOA)-induced plasmid shuffling. Loss of Pbn1 function in GPI biosynthesis was predicted to be lethal, whereas defects in ERAD are compatible with life. Deletion of the complete distal region of GFP-Pbn1 (Δ21–208) did not affect cellular growth, suggesting that this region is not required for GPI biosynthesis (Fig. 6B). The same mutant did not change the MW or degradation rate of ΔNg-CPY*, indicating that this domain is not required for ΔNg-CPY* O-mannosylation or ERAD either (Fig. 6C,D).

Deletion of Pbn1 regions within the conserved Ig-like fold or of the entire TMD resulted in inviability, suggesting that these regions are crucial for GPI biosynthesis (Fig. 6E,F). This finding prevented further analysis of ΔNg-CPY* degradation in these mutants. However, we could isolate a viable mutant where the natural TMD of Pbn1 was replaced with an artificial one composed of seven repeats of the tripeptide LIV. This construct, GFP-Pbn1TMD:LIV, supported growth at a reduced rate, which could suggest that the function in GPI biosynthesis is affected (Fig. 6G,H). Interestingly, this mutant displayed specific ERAD defects, comparable to those seen in cells with depleted Pbn1. Whereas the degradation of ΔNg-CPY* was reduced and paralleled by an accumulation of the subpopulation with lower MW (Fig. 6I, compare lanes 1 and 5, Fig. 6J), the degradation of CPY* remained unaffected, showing that ERAD was not generally impaired or non-functional as a result of reduced cellular growth (Fig. 6K,L). These data further support a role for Pbn1–Gpi14 in protein O-mannosylation and in ERAD.

## Discussion

Our results suggest that protein O-mannosylation promotes ERAD, providing a mechanism for the regulation of protein degradation beyond N-glycan processing. Protein O-mannosylation has previously been shown to terminate futile folding cycles, maintain protein solubility, and promote ER retention, all of which are essential for protein retrotranslocation prior to degradation (Harty et al, 2001; Nakatsukasa et al, 2004; Hirayama et al, 2008; Goder and Melero, 2011; Xu et al, 2013). In summary, our data suggest protein O-mannosylation to be a key modification in ERQC and ERAD.

The identification of the non-canonical bifunctional mannosyl-transferase Pbn1-Gpi14 was central to demonstrating that protein O-mannosylation promotes ERAD in yeast. The dual role of the Pmt1/2 complex—in both O-mannosylation and promoting ER retention—has partially obscured the direct link between O-mannosylation and ERAD. Although the Pmt1/2 complex was shown to associate with the Hrd1 complex in yeast, its deletion did not consistently stabilize misfolded proteins. Instead, it often resulted in accelerated degradation due to increased routing of proteins to the vacuole under such conditions (Fig. 3E,F and Hirayama et al, 2008; Goder and Melero, 2011; Sikorska et al, 2016). In contrast, our data show that reduced protein O-mannosylation due to Pbn1-Gpi14 depletion did not enhance protein turnover and instead led to stabilization of most tested substrates. Based on our functional data, we propose that protein O-mannosylation adds an additional layer of control in the degradation of certain misfolded N-glycosylated proteins, while also functioning as a fail-safe mechanism for the ERAD of misfolded proteins that lack N-glycans. Our findings support both scenarios.

First, SRRs are potential hotspots for protein O-mannosylation and are found in proteins that are usually N-glycosylated as well, such as many GPI-APs in fungi (de Groot et al, 2003; Lommel and Strahl, 2009). Our data suggest that SRRs may be converted into a degron in a protein folding-dependent manner through O-mannosylation and thus constitute specific regions that regulate ERQC and ERAD. In support, we found that misfolded Gas1* and CPY*SRR were both O-mannosylated and degraded, whereas O-mannosylation was strongly reduced or absent in wild-type Gas1 and CPYwtSRR, concomitant with their slower natural turnover. Whereas O-mannosylation of the SRR was mostly Pmt1/2-dependent, we detected a contribution of Pbn1-Gpi14 to this process in all misfolded model proteins containing an SRR used in this study. Our data obtained with SRR-containing CPY* species showed that protein O-mannosylation of an SRR can be extensive. A more than 10 kDa MW increase in CPY*SRR_TMD suggests O-glycan chain formation at individual serines within the SRR domain. Elongation of posttranslational modifications for the regulation of degradation efficiency is a recurring theme. For example, the generation of elongated heterotypic branched ubiquitin chains enhances proteasomal degradation (Meyer and Rape, 2014; Leto et al, 2019).

Second, O-mannosylation and degradation of ΔNg-CPY*, but not CPY*, by Pbn1-Gpi14 suggests that O-mannosylation might function as a fail-safe mechanism for proteins lacking N-glycans. Our observations suggest that protein O-mannosylation of ΔNg-CPY* is dynamic. For example, although conA-bound mannosylated ΔNg-CPY* in Δhrd1 cells contained higher molecular weight species compared to control cells, as expected, a large fraction of the accumulated protein was not stably mannosylated at steady state. Failure of degradation might therefore result in de-mannosylation by ER-resident ER mannosidases, like the processing of N-glycans. Alternatively, Pbn1-Gpi14-mediated protein O-mannosylation efficiency may be coupled to active retrotranslocation.

Both O-mannosylation- and N-glycosylation are glycan-based modifications that affect ERAD, yet they differ fundamentally in how they occur and allow for the regulation of protein degradation in distinct ways. N-glycans are co-translationally attached to proteins, with a defined number and location on each protein. Through time-dependent enzymatic trimming, they can generate a localized set of ERAD cues. We propose that O-mannoses may also serve as ERAD cues, distinct from canonical N-glycan signals. O-mannoses are attached post-translationally, on demand, and in potentially variable quantities, offering flexibility in their placement on proteins due to their ability to form on virtually any accessible hydroxyl-containing amino acid. Such a mechanism could enable broader control over protein degradation, regulating both timing and kinetics of the process.

Our results suggest that PMTs and Pbn1-Gpi14 exhibit distinct specificities towards the utilized model substrates. The ERQC-relevant Pmt1/2 complex efficiently O-mannosylated the unstructured flexible SRR on substrates like Gas1* and CPY*SRR_TMD, thereby influencing their ERAD, while ΔNg-CPY*, which only contains scattered hydroxylated amino acids, remained unmodified by this complex. Previous studies suggested that scattered hydroxyl amino acids are not per se targets for O-mannosylation by the Pmt1/2 complex (Lommel and Strahl, 2009; Chiapparino et al, 2020). Conversely, Pbn1-Gpi14 was less efficient in modifying the tested SRR but was essential for the O-mannosylation of ΔNg-CPY*. The substrate specificities of cellular protein O-mannosyltransferases may also be influenced by cellular conditions. For instance, ER stress was suggested to enhance PMTs' affinity for substrates with scattered serines or threonines by increasing their concentration at the ER membrane (Hutzler et al, 2007).

In yeast, GPI- and protein-O-mannosyltransferases share functional roles and evolutionary origins, as indicated by our findings. For instance, our phylogenetic analysis linked Gpi14 with PMTs. Protein O-mannosylation by Pbn1-Gpi14 may represent an ancestral mechanism for protein modification, predating the evolutionary emergence of specialized PMTs with MIR domains. We also found that overexpression of Pbn1 accelerated the degradation rates of the tested misfolded SRR-containing proteins. Meanwhile, Gpi14 overexpression influences mannose conjugation to the GPI precursor in Leishmania parasites (Ribeiro et al, 2019). Both findings support a bifunctionality of Pbn1-Gpi14. Unlike Pbn1-Gpi14, protein-specific PMTs are UPR-controlled, exhibit high SRR affinity, and contribute to ER retention. This might reflect the speciation of PMTs for exclusive modification of proteins.

Our data linking Pbn1 to protein O-mannosylation also suggest an updated model for the initially proposed chaperone functions of Pbn1 in yeast, whose mechanism of action has remained elusive. Pbn1 was originally identified for its role in the autocatalytic processing of the subtilisin-like vacuolar yeast protease Prb1 within the ER (Naik and Jones, 1998). Although previous models suggested that Pbn1 serves as a scaffold for Prb1 self-cleavage, our findings would suggest a role of Pbn1 in the observed O-mannosylation of the Prb1 precursor and the subsequent degradation of its large 39 kDa cleavage product (Moehle et al, 1989; Naik and Jones, 1998).

Based on our findings, we propose a model in which protein O-mannosylation generates active degradation signals (ERAD cues). O-linked mannoses share structural similarities with terminal mannoses on processed N-glycans and may recruit similar or identical ERAD-associated lectins. We further suggest that the previously proposed roles of O-mannosylation in maintaining ER homeostasis—such as preventing protein aggregation caused by impaired N-glycosylation and terminating futile folding cycles of terminally misfolded proteins—are part of the same overarching process. Identifying ER-localized receptors for O-linked mannoses in the context of ERAD would provide further evidence for an active role of protein O-mannosylation in this pathway, extending beyond its earlier characterization as a passive "enabler" of degradation.

# Methods

**Reagents and tools table**

| Reagent/resource | Reference or source | Identifier or catalog number |
|---|---|---|
| **Experimental models** | | |
| *Saccharomyces cerevisiae* laboratory yeast strains | This study | Appendix Table S1 |
| **Recombinant DNA** | | |
| Plasmids | This study | Appendix Table S2 |
| **Antibodies** | | |
| Anti-GFP | Roche | 11814460001; RRID:AB_390913 |
| Anti-HA (12CA5) | Roche | 11666606001 ROAHA |
| Anti-HA High Affinity (3F10) | Roche | 11867423001, RRID: AB_390918) |
| Anti-mouse-HRP | Sigma-Aldrich | A9044; RRID:AB_258431 |
| Anti-MYC | Roche | 11667149001; RRID:AB_390912 |
| Goat Anti-Mouse light chain Antibody, HRP conjugate | Sigma-Aldrich | AP200P |
| Goat Anti-Rat light chain Antibody, HRP conjugate | Sigma-Aldrich | AP202P |
| **Chemicals, enzymes, and other reagents** | | |
| Western ECL Substrate | Thermo Scientific Pierce | 10590624 |

| Reagent/resource | Reference or source | Identifier or catalog number |
|---|---|---|
| cOmplete™, Mini, EDTA-free Protease Inhibitor Cocktail | Roche | 11836170001 |
| Con A Sepharose 4B | Amersham Biosciences | 17044001 |
| CSM-HIS-LEU-TRP-URA, powder amino acid supplement | MP Biomedicals | 114540012-CF |
| Cycloheximide | Sigma-Aldrich | C7698 |
| Dimethyl sulfoxide (DMSO) | Sigma-Aldrich | D8418 |
| Dithiothreitol (DTT) | Sigma-Aldrich | D5545 |
| Geneticin | Calbiochem | 345810 |
| HygromycinB | Invitrogen | 10687010 |
| Nourseothricin | Jena Bioscience | AB-102 |
| NP-40 Surfact-Amps™ Detergent Solution | Thermo Fisher Scientific | 85124 |
| PNGase F | New England Biolabs | P0704S |
| SuperSignal West Pico PLUS Substrate | Thermo Fisher Scientific | 34577 |
| T4 DNA Ligase | New England Biolabs | M0202T |
| Thialysine | Sigma-Aldrich | A2636 |
| Tunicamycin | Sigma-Aldrich | T7765 |
| Yeast Nitrogen Base without Amino Acids and Ammonium Sulfate | Fisher Scientific | 11743014 |
| α-Mannosidase from *Canavalia ensiformis* (Jack bean) | Sigma-Aldrich | M7257 |
| **Software** | | |
| AlphaFold 3 | https://alphafoldserver.com/welcome | |
| ChimeraX (version 1.7) | https://www.rbvi.ucsf.edu/chimerax/ | |
| GraphPad Prisma 10 | https://www.graphpad.com/ | |
| Image Studio software | https://www.licorbio.com/image-studio | |
| ImageJ/FIJI | https://imagej.net/ij/ | |
| PyMOL | https://www.pymol.org/ | |
| PhyML | https://ngphylogeny.fr/ | |
| **Other** | | |
| ChemiDoc™ MP Imaging System | Bio-Rad | |
| Corning® Falcon® Cell Culture Plate | Sigma-Aldrich | CLS353226 |
| FastPrep-24™ 5 G Bead Beating Grinder. MP Biomedicals™ | Fisher Scientific | 15260488 |
| FLUOstar Omega Microplate Reader | BMG Labtech | 3374219 |
| inverted fluorescent microscope ScanR High-Content Screening | Olympus | |

| Reagent/resource | Reference or source | Identifier or catalog number |
|---|---|---|
| ODYSSEY® XF Imaging System | LICORbio | |
| RoToR HDA | Singer Instruments | |
| Trans-Blot® Turbo™ Transfer System | Bio-Rad | |

## Yeast strains used in this study

All experiments were performed using common laboratory yeast strains constructed in a BY (MATa/α *his3Δ1/his3Δ1 leu2Δ0/leu2Δ0 LYS2/lys2Δ0 met15Δ0/MET15 ura3Δ0/ura3Δ0*) or W303 (*leu2-3,112 trp1-1 can1-100 ura3-1 ade2-1 his3-11,15*) background. Genomic tagging of yeast strains was performed using standard PCR-based amplifications from suitable plasmids followed by transformation with PCR products and PCR-based verifications and/or western blot analysis. A list of yeast strains used in this study is found in Appendix Table S1. All newly generated yeast strains are available upon request.

## Plasmids used in this study

A list of plasmids used in this study is found in Appendix Table S2. All plasmid-borne constructs were generated by standard PCR-based amplifications and/or restriction enzyme-based subcloning. All relevant constructs have been verified by sequencing. All newly generated plasmids are available upon request.

## Genome-wide yeast screens

Screen using Leu2-TMD-Gas1* as a reporter. The yeast knock-out (YKO) deletion collection and the *DAmP* collection were grown to logarithmic growth phase in 96-well plates containing rich medium and transformed with plasmid VGp161, containing Leu2-TMD-Gas1*. Transformants were selected by growth in 96-well plates containing synthetic media lacking uracil. Transformants were transferred to 96-well plates containing synthetic media lacking uracil and, in parallel, to 96-well plates containing synthetic media lacking both uracil and leucine. Plates were incubated at 30 °C and evaluated for growth after five days. Transformants with growth in both media were designated as hits and are listed in Dataset EV1. Screen using GFP-Gas1* as a reporter: GFP-tagged. BY4741 containing a *Δted1* deletion was used for genomic integration of GFP-Gas1* (VGc226) for expression from its endogenous promoter. The strain was crossed with the collection of yeast strains expressing N-terminal mCherry fusions to all individual ORFs under the control of the strong constitutive *TEF2* promoter (Weill et al, 2018). For automated microscopy, cells were transferred from agar plates into 384-well polystyrene plates (Greiner) for growth in liquid media using the RoToR HDA robot (Singer Instruments). Liquid cultures were grown in a shaking incubator (Liconic), overnight at 30 °C. A Freedom EVO 200 liquid handler (Tecan) connected to the incubator was used to prepare the cells for microscopy by transferring them into glass-bottom 384-well microscope plates (Matrical Bioscience) coated with Concanavalin A (Sigma-Aldrich). After 20 min, cells were washed three times in

complete medium to remove non-adherent cells and to obtain a cell monolayer. The plates were then transferred to an automated inverted fluorescent microscope ScanR High-Content Screening system (Olympus) harboring a spinning disk module (Yokogawa CSU-W1 spinning disk confocal scanner with 50 μm pinhole) using a robotic swap arm. Images of cells in the 384-well plates were recorded in the same liquid as the washing step at 24 °C using a ×60 air lens (NA 0.9) and with a Hamamatsu ORCA-Flash 4.0 camera. Images of GFP (wavelength of 488 nm) were acquired following excitation by a laser and using an emission filter of 513/30 nm). Automated thresholding was performed to identify hits that showed >20% signal reduction compared to the median GFP signal. Hits are listed in Dataset EV1.

## Plating assays

Cells were grown in the indicated liquid medium overnight at 30 °C, adjusted to the same optical density and spotted with identical volumes in 1:10 dilution series onto plates containing rich or synthetic medium lacking the indicated components. Plates containing 5-FOA (5-fluoroorotic acid) were used to select for cells that have lost the *URA3*-containing plasmid together with the gene of interest. *URA3* encodes the enzyme orotidine-5'-monophosphate decarboxylase, which, in the presence of 5-FOA, is converted to toxic 5-fluorouracil. Plates were incubated for the indicated times and temperatures prior to imaging.

## Cycloheximide (CHX) shut-off experiments

The experiments were started with exponentially growing cells at 30 °C in a synthetic medium with an absorbance at 600 nm between 0.5 and 0.8. Translation was stopped by the addition of CHX to a final concentration of 200 μg/ml. Equal volume aliquots of cell culture were removed at indicated time points and moved to ice. Cells were pelleted, resuspended in 500 μl cold 150 mM NaOH and left on ice for five minutes, centrifuged and lysed by adding sample buffer containing 2% SDS and heating to 65 °C for 10 min.

## Treatment of cells with tunicamycin

Tunicamycin (SIGMA) dissolved in DMSO was applied to exponentially growing cells to a final concentration of 1 μg/ml, or to the concentration indicated. Control cells were treated with the same volume of DMSO (solvent). Cells were lysed after the indicated times and processed like non-treated cells for subsequent analyses.

## SDS-PAGE and western blotting

Samples were analyzed by standard SDS-PAGE followed by western blotting using the indicated primary antibodies, peroxidase-coupled secondary antibodies (Roche) and enhanced chemiluminescence (ECL, Thermo Fisher Scientific) as substrate. Images were taken with a LICOR imaging system and bands were quantified using Image Studio software (LICOR). For time course experiments, significance was calculated using two-way ANOVA with Šidák's correction for multiple comparisons. All statistical analyses were performed in GraphPad Prism 10 software.

## Treatment with PNGaseF

Exponentially growing cells were lysed in cold phosphate-buffered saline pH 7.4 containing protease inhibitor cocktail (Roche) using a FastPrep homogenizer (MP Biomedicals) in combination with glass beads and bead beating. The lysate was transferred and SDS was added to a final concentration of 0.5%. Lysate was heated to 95 °C for 10 min, vortexed briefly and transferred to a new tube. Prior to the addition of PNGase F (New England Biolabs), G7 buffer and NP40 (both New England Biolabs) were added according to the manufacturer's instructions. Lysate supplemented with PNGase F and control lysate were incubated for 2 h at 37 °C. The reaction was terminated by the addition of sample buffer containing 2% SDS and heating to 65 °C for 20 min.

## Treatment with α-mannosidase from *Canavalia ensiformis* (Jack bean)

The reaction was performed according to the manufacturer's instructions. In brief, cells were pelleted, washed, and resuspended in sodium acetate buffer pH 4.5, together with 2 mM $ZnCl_2$ and protease inhibitor cocktail (Roche). Cells were lysed using glass beads and bead beating with a FastPrep homogenizer (MP Biomedicals). The lysate was transferred and SDS was added to a final concentration of 0.5%. Lysate was heated to 95 °C for 10 min, vortexed briefly, and transferred to a new tube. NP-40 was added to a final concentration of 1%. Sample was split into two equal parts. In all, 5 μl of a 3 M ammonium sulfate solution containing jack bean mannosidase (SIGMA) was added to one sample and the same amount of 3 M ammonium sulfate to the control sample. Samples were incubated at 37 °C for 4 h, with stirring at 500 rpm. The reaction was terminated with the addition of sample buffer containing 2% SDS and heating to 65 °C for 20 min.

## Precipitation of glycosylated protein with conA-Sepharose

Exponentially growing cells were pelleted, moved to ice and resuspended in 1 ml 150 mM NaOH, left for 5 min on ice and pelleted. For lysis, cells were resuspended in 150 μl phosphate-buffered saline pH 7.4, containing protease inhibitor cocktail (Roche), 2% SDS, and 10 mM dithiothreitol and incubated at 95 °C for 10 min. Cells were briefly vortexed, pelleted and the supernatant was transferred to a new tube. An aliquot was removed for "input" and to the rest 1250 μl phosphate-buffered saline pH 7.4, containing protease inhibitor cocktail (Roche), 100 μl washed concanavaline-Sepharaose beads (Amersham), and 1% bovine serum albumin (BSA) was added. The lysate was incubated rotating at 4 °C for 3 h. Beads were pelleted and washed four times with phosphate-buffered saline pH 7.4, containing protease inhibitor cocktail (Roche) and 0.1% SDS. Bound material was eluted by incubation with sample buffer containing 2% SDS and heating to 65 °C for 20 min.

## Growth assays

Cells were grown in the indicated liquid medium overnight at 30 °C, reset to an optical density (OD600 nm) of 0.05 in a volume of 500 μl in a 24-well plate in duplicates and incubated for 24 h in a

BMG FLUOstar Omega multi-mode microplate reader at 700 rpm and a temperature of 30 °C. Indicated drugs or only solvents were applied at the indicated times. ODs were recorded automatically every hour.

## GFP-processing assays

Cells were grown overnight, diluted to OD 0.2 and regrown for 5 h. Removed aliquots were lysed by alkaline treatment (Kushnirov, 2000), resuspended in a cell-density normalized volume of loading buffer, followed by SDS-PAGE and western blotting using anti-GFP antibody (Roche), HRP-conjugated anti-mouse secondary antibody (Roche), and ECL (PIERCE) as substrate. Images were taken with a LAS-3000 mini imaging system (Fujifilm) and bands were quantified using Multi-Gauge software (Fujifilm).

## Fluorescence microscopy

Exponentially growing cells were washed with PBS and immediately analyzed by fluorescence microscopy. Cells were observed with a LEICA DMi8 microscope equipped with a ×100/1.4 oil Plan-Apo immersion lens and a DIC prism and polarizer for Normarksi imaging. Images were acquired using a Hamamatsu C1 3440-20CV camera and the LASX controller software (Leica).

## Confocal live-cell microscopy

Exponentially growing cells were washed with PBS and scanned with a Laser Scanning Confocal Microscope from Zeiss (LSM 7 Duo) equipped with a BiG (binary GaAsP) module using a Plan-Apochromat objective ×63/1.40 Oil DIC. A 488 nm Argon Laser (GFP) and a 561 nm Helium-Neon Laser were used (tdimer, mCherry).

## Structure predictions and structure-based search

Superpositioning of AlphaFold structure predictions was done with PyMOL using the align function. The superimposed structures were visualized using PyMOL's graphical user interface. Protein complex prediction was performed using AlphaFold3 or AlphaFold Multimer in combination with ChimeraX (version 1.7). Special effects for visualization of complexes were obtained with ChimeraX using a combination of ribbon and surface representations. The individual proteins were colored distinctly for clarity. Specific residues involved in catalytic activity were highlighted and labeled. Structure-based search was performed using Foldseek (van Kempen et al, 2024).

## Phylogenetic analysis

The analysis was performed according to instructions (https://ngphylogeny.fr/). All amino acid sequences were provided in FASTA format, and the fully automated workflow with PhyML as tree inference was selected. This included the following steps and default parameters: Multiple Alignment (MAFFT: flavor: auto; Gap extension penalty: 0.123; Gap opening penalty: 1.53), Alignment Curation (BMGE: Sliding window size: 3; Maximum entropy threshold: 0.5; Gap rate cutoff: 0.5; Minimum block size: 5; Matrix: BLOSUM62), Tree inference (PhyML: Statistical criterion to select

the model: AIC; Tree topology search: SPR; Branch support: No branch support; Proportion of invariant sites: Estimated; Number of categories for the discrete gamma model: 4; Parameters of the gamma model: estimated; Tree topology search: SPR; Optimize parameters: tlr; Model: LG; Equilibrium frequencies: ML model) and Tree Rendering (Newick Display: Model: LG; Equilibrium frequencies: estimated; Gamma distributed rates across sites: Yes; Gamma distribution parameter: 1.0; Remove gap strategy: Pairwise deletion of gaps; Starting tree: BioNJ; Tree Refinement: BalME SPR; Bootstrap: No; Decimal precision for branch lengths: 6).

## Data availability

This study includes no data deposited in external repositories.

The source data of this paper are collected in the following database record: biostudies:S-SCDT-10_1038-S44318-025-00647-2.

## Peer review information

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

## Acknowledgements

We thank Piet de Groot for advice, Yoshifumi Jigami, Tom Rapoport, Pedro Carvalho, and Jonathan Weissman for reagents. We gratefully acknowledge Michael Van de Weijer for assistance with GraphPad Prism. We further thank Pedro Carvalho and Sabine Strahl for their critical reading of the manuscript and for many helpful comments. We thank former members of the Goder lab: Enrique Alanis, Mar Bustamante, and Carlos Ruiz for technical support. MS was supported by grants from the EU-H2020-ERC-CoG (OnTarget 864068). The robotic system of the Schuldiner lab was purchased through the kind support of the Blythe Brenden-Mann Foundation. MS is an Incumbent of the Dr. Gilbert Omenn and Martha Darling Professorial Chair in Molecular Genetics. LL and VG were supported by a grant from the Agencia Estatal de Investigación (AEI/10.13039/501100011033/PID2022-136665NB-I00). LL was supported by the "Talento Doctor" Fellowship (co-funded by the European Regional Development Fund (ERDF) and the Junta de Andalucía, Spain), and by the EMBO Scientific Exchange Grant (11267) for a research visit to the laboratory of Pedro Carvalho at the Sir William Dunn School of Pathology, University of Oxford, UK.

## Author contributions

**Leticia Lemus**: Conceptualization; Data curation; Software; Formal analysis; Supervision; Funding acquisition; Validation; Investigation; Methodology; Writing—original draft; Writing—review and editing. **Hadar Meyer**: Data curation; Software; Validation; Investigation; Methodology. **Ana I Rodríguez-Rosado**: Data curation; Validation; Investigation; Methodology. **Maya Schuldiner**: Conceptualization; Data curation; Supervision; Funding acquisition; Validation; Investigation; Methodology; Writing—review and editing. **Veit Goder**: Conceptualization; Data curation; Formal analysis; Supervision; Funding acquisition; Validation; Investigation; Methodology; Writing—original draft; Project administration; Writing—review and editing.

Source data underlying figure panels in this paper may have individual authorship assigned. Where available, figure panel/source data authorship is listed in the following database record: biostudies:S-SCDT-10_1038-S44318-025-00647-2.

## Disclosure and competing interests statement

The authors declare no competing interests.

# Expanded View Figures

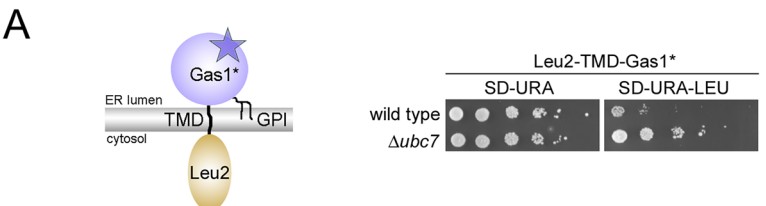

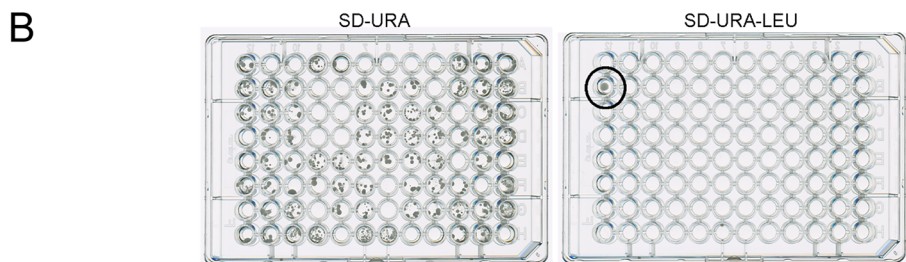

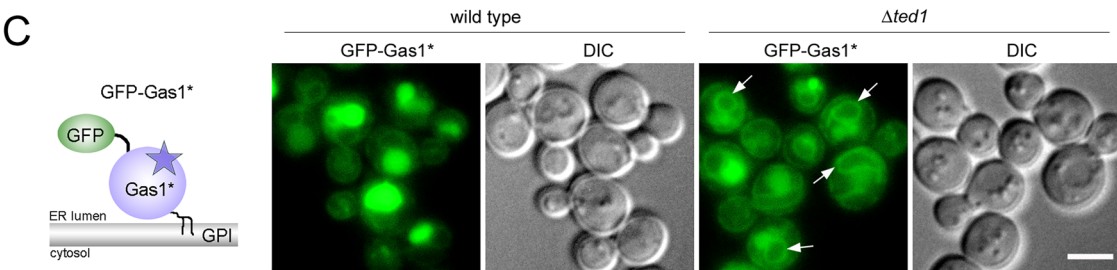

**Figure EV1.  Setup of complementary genome-wide screens.**

(**A**) Design and functionality test of the first screen. Wild-type cells and cells with *UBC7* deleted (*Δubc7*) were used to express the plasmid-borne reporter construct Leu2-TMD-Gas1* (graphical depiction). The plasmid contained *URA3* as a marker and expressed the reporter from the *GAL4* promoter. Cells were grown in medium lacking uracil and spotted in serial dilutions in equal concentration on plates containing synthetic medium lacking uracil (SD-URA) or lacking both uracil and leucine (SD-URA-LEU). Plates were incubated for 3 days (SD-URA) or 5 days (SD-URA-LEU) at 30 °C prior to imaging. The growth of *Δubc7* mutant cells in the absence of leucine (SD-URA-LEU) indicated an impairment in reporter degradation, as expected. (**B**) Screen readout. Representative example for pairs of 96-well plates used for the identification of hits. Transformants of the yeast deletion collection (*Δxxx*) and the collection of decreased abundance by mRNA perturbation (*DAmP*) alleles were first grown in 96-well plates containing liquid synthetic medium lacking uracil (SD-URA). Note that the organization of the libraries is such that not all wells contain cells. After growth in SD-URA, cells were replica-plated into 96-well plates containing medium lacking uracil (SD-URA) and lacking both uracil and leucine (SD-URA-LEU), a pair of plates is shown as example. Growth in both media was considered a hit (circled well). (**C**) Design of the second screen. Wild-type cells and cells with *TED1* deleted (*Δted1*) were used to express genomic N-terminally GFP-tagged Gas1* (graphical depiction) for live-cell high-throughput fluorescence microscopy and differential interference contrast (DIC) microscopy. Ted1 is involved in GPI anchor remodeling. Its deletion was known to reduce the ER export and vacuolar targeting of Gas1* and to increase its routing to ERAD (Sikorska et al, 2016). In agreement, while most GFP was visible inside vacuoles in wild-type cells, indicating efficient targeting of GFP-Gas1* to vacuoles, a significant fraction of the protein accumulated in the perinuclear ER in *Δted1* cells (arrows). Differential interference contrast (DIC) microscopy was used to identify vacuoles. Scale bar: 4 μm.

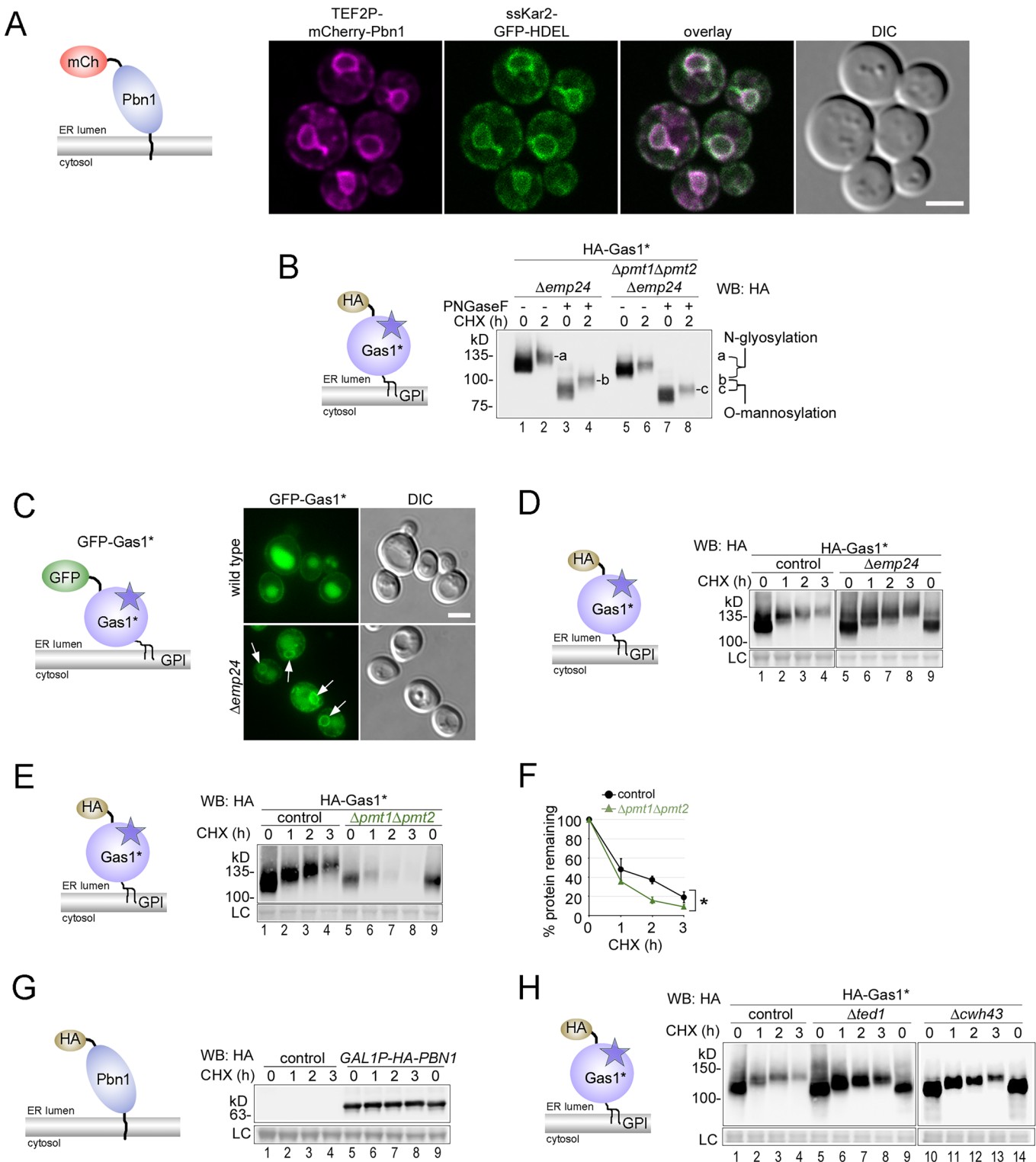

**Figure EV2.  Cellular localization of Pbn1 and Gas1\*.**

Changes in the O-mannosylation of Gas1\* in different mutants. (A) Pbn1 localizes to the ER. Yeast cells co-expressing genomic N-terminally mCherry-tagged Pbn1 from the strong constitutive *TEF2* promoter (graphical depiction) and the ER marker ssKar2-GFP-HDEL were analyzed by live-cell fluorescence and DIC microscopy. The degree of colocalization indicated that Pbn1 does not "leak" from the ER, even at a high expression level. Scale bar: 3 μm. (B) Gas1\* is both N-glycosylated and O-mannosylated. Cells with the indicated deletions were used to express plasmid-borne HA-tagged Gas1\* (graphical depiction) for cycloheximide (CHX) shut-off experiments. To prevent rapid ER export and vacuolar degradation of Gas1\* caused by deletion of *PMT1* and *PMT2* (*Δpmt1Δpmt2*), the experiment was performed with an additional mutation, in the *Δemp24* background to retain the substrate in the ER, as shown in the fluorescence image ((Fig. EV2C and Goder and Melero, 2011). Cells were lysed at the indicated time points and, where marked (+), lysates were treated with Peptide-N-Glycosidase F (PNGaseF) for the removal of all N-linked glycans, prior to SDS-PAGE and western blot (WB) analysis with anti-HA antibodies. The visible shift in MW after treatment with PNGaseF was reflective of extensive Gas1\* N-glycosylation (compare lanes 2 and 4). The additional shift in MW visible in *Δpmt1Δpmt2Δemp24* cells compared to *Δemp24* cells was indicative of protein O-mannosylation (compare lanes 4 and 8). (C) Genetic blocking of Gas1\* ER export. Wild-type cells and *Δemp24* cells were used to express genomic N-terminally GFP-tagged Gas1\* (graphical depiction) for live-cell fluorescence microscopy and DIC microscopy. The fraction of GFP-Gas1\* that is targeted to the vacuole is visible due to the accumulation of vacuolar GFP. Deletion of *EMP24* is known to block ER export of Gas1\* and to reduce the protein population that is targeted to the vacuole (Sikorska et al, 2016). Arrows indicate accumulated GFP-Gas1\* in the ER in *Δemp24* cells. Scale bar: 3 μm. (D) Gas1\* O-mannosylation occurs inside the ER. Control cells and *Δemp24* cells were used to express plasmid-borne HA-tagged Gas1\* for CHX shut-off experiments. Cells were lysed at the indicated time points after the addition of CHX, and the remaining HA-Gas1\* was measured by SDS-PAGE and WB with anti-HA antibodies. Membrane staining with Ponceau served as a loading control (LC). The comparable increase in molecular weight (MW) of Gas1\* prior to degradation in both control cells and *Δemp24* cells, despite genetically blocked ER exit in the latter, is indicative of extensive protein O-mannosylation occurring within the ER. Stabilization of Gas1\* in *Δemp24* cells was reported previously (Sikorska et al, 2016). (E) Gas1\* degradation is accelerated in the absence of the Pmt1/2-complex. Control cells and *Δpmt1Δpmt2* cells were used to express plasmid-borne HA-tagged Gas1\* for CHX shut-off experiments. Cells with *PMT1* and *PMT2* deleted (*Δpmt1Δpmt2*) showed two effects. First, the increase in MW was drastically reduced (compare lanes 1–4 with 5–9), in agreement with the protein being O-mannosylated by the Pmt1/2-complex. Second, Gas1\* was degraded faster (compare lanes 1–4 with 5–9, and graph). Faster degradation was previously reported to be caused by a loss of ER retention and increased routing to the vacuole under these conditions (Goder and Melero, 2011; Sikorska et al, 2016). (F) Quantification of results shown in (E). (G) Stability of HA-tagged Pbn1 expressed from the *GAL1* promoter. Control cells and cells expressing genomic N-terminally HA-tagged Pbn1 from the *GAL1* promoter (graphical depiction) were used for CHX shut-off experiments. Cells were grown in medium containing galactose as a carbon source. Overexpressed Pbn1 remained stable throughout the experimental period. This strain background was used for all experiments using the overexpression of HA-Pbn1. The loading control is identical to that shown in Fig. 2E, as different sections of the same membrane were used for both figures. (H) No non-O-mannosylated Gas1\* was detectable in deletion mutants that impact GPI metabolism. Control cells and cells with the indicated deletions were used to express plasmid-borne HA-tagged Gas1\* for CHX shut-off experiment and processed like in (D). Stabilization of Gas1\* in these mutants is due to blockage of ER export and was reported previously (Sikorska et al, 2016). Data information: Number of experiments (*n*) signifies biological replicates. Error bars in graphs represent the standard deviation from the mean. Statistical significance for experiments involving degradation rates was calculated using two-way ANOVA (alpha = 0.05) with Šidák's correction for multiple comparisons, obtaining the following *p* values. (F): (*n* = 2, *p* = 0.0103). Difference in degradation rate: \**p* < 0.05.

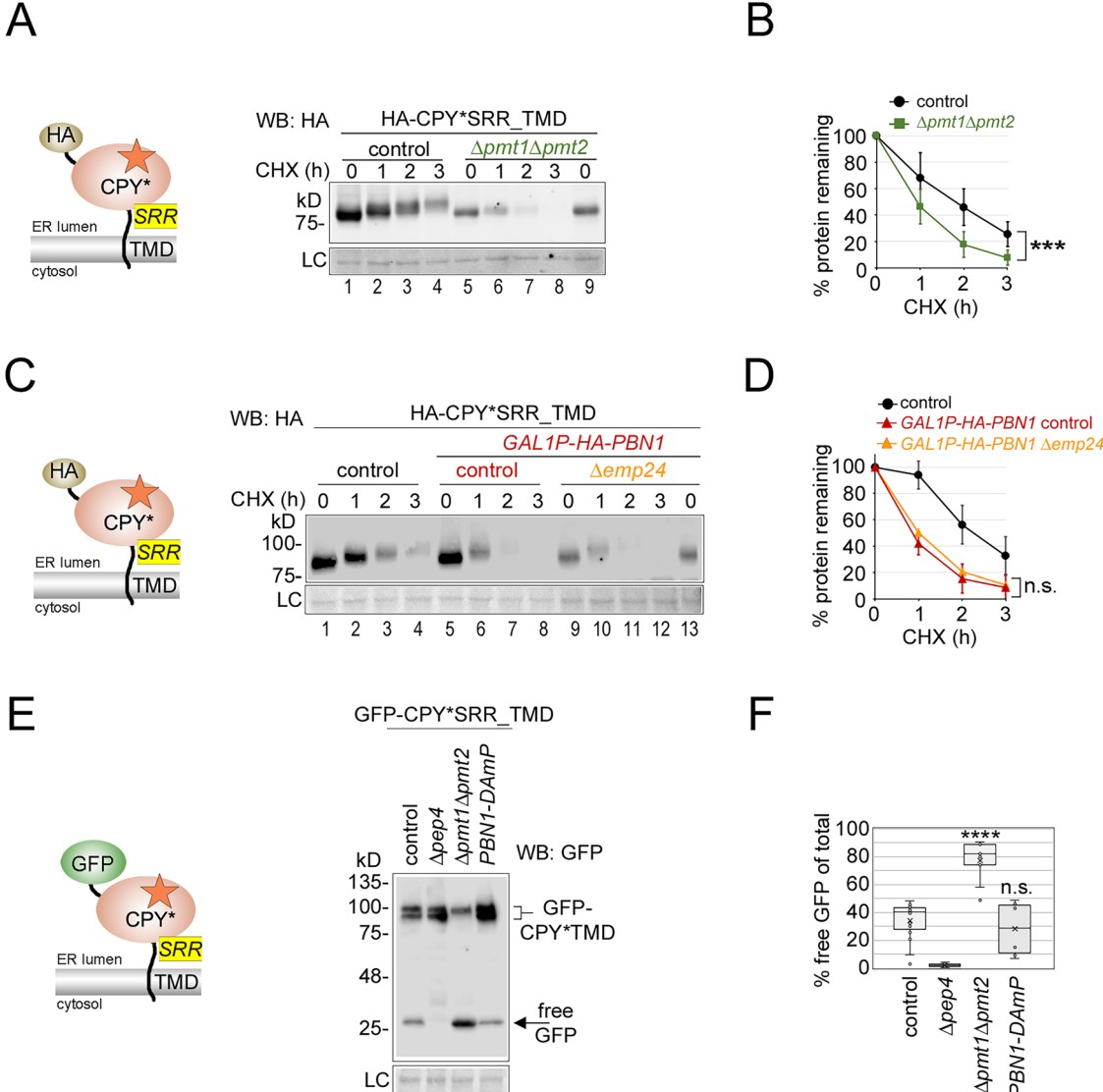

**Figure EV3. Characterization of CPY*SRR_TMD.**

(A) CPY*SRR_TMD is O-mannosylated by the Pmt1/2-complex. Control cells and cells with *PMT1* and *PMT2* deleted (*ΔpmtΔpmt2*) were used to express plasmid-borne HA-tagged CPY*SRR_TMD (graphically depicted) for CHX shut-off experiments. Cells were lysed at the indicated time points after the addition of CHX, and the remaining protein was measured by SDS-PAGE and WB with anti-HA antibodies. As seen with Gas1*, the increase in MW of CPY*SRR_TMD was reduced in *ΔpmtΔpmt2* cells, indicating that the protein was O-mannosylated by the Pmt1/2-complex (compare lanes 1–4 with 5–9). (B) Quantification of results shown in (A). (C, D) CPY*SRR_TMD degradation kinetics was unaffected in *Δemp24* cells upon Pbn1 overexpression. Control cells and cells containing *GAL1P-HA-PBN1* and lacking *EMP24* (*Δemp24*) where indicated were used to express plasmid-borne HA-CPY*SRR_TMD (depicted). Cells were grown overnight in medium containing raffinose and then diluted into medium containing galactose for the expression of HA-Pbn1 three hours prior to the application of CHX. Cells were processed as described in (A). (E) Increased routing of CPY*SRR_TMD to the vacuole in *Δpmt1Δpmt2* cells but not in *PBN1-DAmP* cells. To test the amount of CPY*SRR_TMD routed to the vacuole in different backgrounds, we utilized the well-known GFP-cleavage assay (Klionsky et al, 2021). A GFP-tagged version of CPY*SRR_TMD was generated (graphical depiction), leading to the appearance of (free) GFP in WB analysis after vacuolar degradation of the protein, due to the proteolytic stability of the GFP tag (lane 1, "free GFP"). Free GFP was generated inside the vacuole because it was not generated in the absence of the vacuolar master protease Pep4 (lane 2). The relative amount of free GFP was increased in *Δpmt1Δpmt2* cells, indicative of increased routing of CPY*SRR_TMD to the vacuole (lane 3). These findings support previous data that PMTs promote ER retention in addition to protein O-mannosylation (Goder and Melero, 2011; Sikorska et al, 2016). Cells carrying the *PBN1-DAmP* allele did not show increased routing to the vacuole (lane 4). These results suggest that Pbn1 does not possess ER retention activity. (F) Quantification of results shown in (E). Box plot displaying the percentage of free GFP relative to the total GFP signal in each lysate. Data information: Number of experiments (*n*) signifies biological replicates. Error bars in graphs represent the standard deviation from the mean. Statistical significance for all experiments involving degradation rates was calculated using two-way ANOVA (alpha $= 0.05$) with Šidák's correction for multiple comparisons, obtaining the following *p* values. (B): ($n = 9$, $p = 0.0006$), ***$p < 0.001$; (D): ($n = 2$, $p = 0.9998$), n.s. $=$ not significant. (F) Statistical significance of changes of free GFP in *Δpmt1Δpmt2* and *PBN1-DAmP* cells compared to control cells was determined using an unpaired two-tailed Student's *t* test. *Δpmt1Δpmt2*: $n = 9$, $p = 2.96 \times 10^{-7}$; *PBN1-DAmP*: $n = 6$, $p = 0.4373$. n.s. $=$ not significant; ****$p < 0.0001$. Numerical values of the box plot shown in (F): control: (minimum: 3.1; maximum: 48.3; median: 40.5; Q1: 27.9; Q3: 43.3; lower whisker: 9.9; upper whisker: 48.3); *Δpep4*: (minimum: 0.4; maximum: 4.5; median: 2.1; Q1: 1.3; Q3: 3.2; lower whisker: 0.4; upper whisker: 4.5); *Δpmt1Δpmt2*: (minimum: 49; maximum: 91; median: 82; Q1: 74; Q3: 89; lower whisker: 58; upper whisker: 91); *PBN1-DAmP*: (minimum: 7; maximum: 49; median: 29; Q1: 11; Q3: 45; lower whisker: 7; upper whisker: 49).

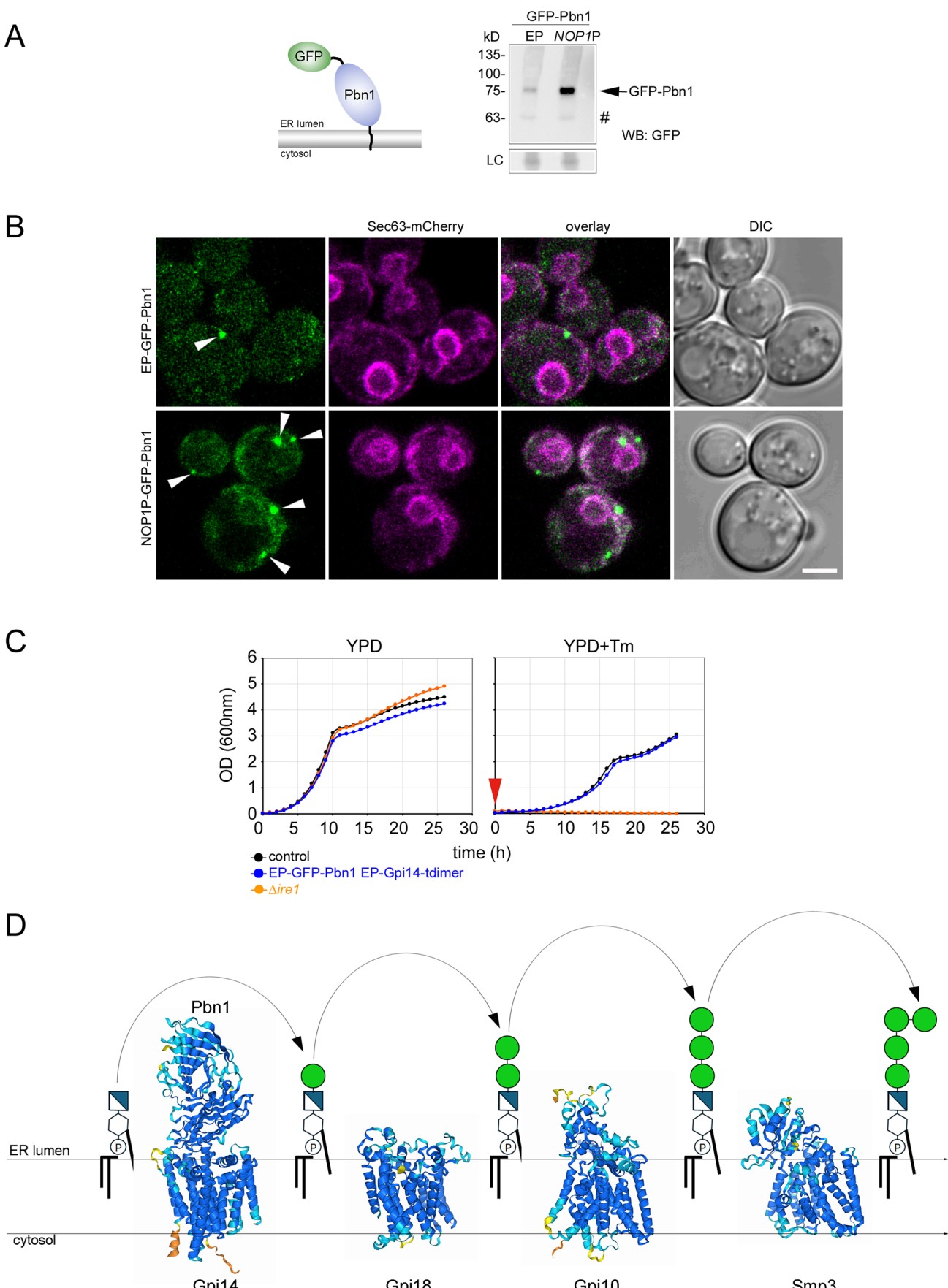

◀

**Figure EV4.  Properties of Pbn1 and Gpi14.**

(A) Comparison of cellular levels of GFP-tagged Pbn1 expressed from different promoters. Expression of genomic N-terminally GFP-tagged Pbn1 (graphical depiction) from either the endogenous promoter (EP) or from the moderate *NOP1* promoter (NOP1P). Equal amounts of cells were taken for lysis, and expression levels were compared by SDS-PAGE followed by WB analysis with antibodies against GFP. Hashtag indicates an unspecific band. The results indicated weak expression of GFP-Pbn1 from its endogenous promoter. (B) Cellular localization of GFP-Pbn1. Cells co-expressing genomic GFP-Pbn1 from different promoters like in (A) and the ER marker Sec63-mCherry were analyzed by live-cell confocal fluorescence microscopy in combination with DIC microscopy. Note that N-terminal tagging of Pbn1 with GFP led to the concentration of the protein in cellular puncta (arrowheads) regardless of protein expression levels. Co-localization with the ER marker Sec63 suggested that Pbn1 puncta localized to the ER. Scale bar: 2 μm. (C) Functionality tests of tagged versions of Pbn1 and Gpi14. Wild-type cells (control) and strains expressing genome-integrated fluorescently tagged Pbn1 and Gpi14 were grown in complete synthetic media (SD) in the absence or presence of 1 μg/ml tunicamycin (Tm) at 30 °C. The drug was applied immediately after starting the experiment by resetting cellular density (red arrowhead). Growth was determined by automated measuring of the absorbance of the individual cell cultures at 600 nm over a period of 25 h. Cells with *IRE1* deleted (*Δire1*) are known to be sensitive to the induction of ER stress with tunicamycin and were used as a control. The strains co-expressing GFP-Pbn1 and Gpi14-tdimer showed no growth defect in SD medium in the absence or presence of tunicamycin compared to the control strain. Since both proteins, Pbn1 and Gpi14, are essential genes, these results indicate that the utilized tagging did not compromise the essential functions of either protein. (D) Schematic illustrating the four different steps of mannosylation of the GPI precursor and the enzymes involved. The membrane-embedded precursor GlcN-(acyl)PI is mannosylated (green filled circles) in a series of reactions involving the enzymes Pbn1-Gpi14, Gpi18, Gpi10, and Smp3, as depicted. The enzyme structure predictions were obtained with Alphafold3. Additional modifications, such as the addition of ethanolamine phosphate, are not shown. The substrates are not drawn to scale in relation to the enzymes. Note that Pbn1 provides an unusually large ER luminal domain to Gpi14, whereas the other enzymes either lack (Gpi18) or have considerably smaller ER luminal domains (Gpi10 and Smp3), although a similar GPI precursor structure is modified in each case.

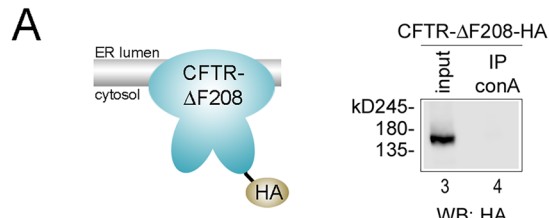

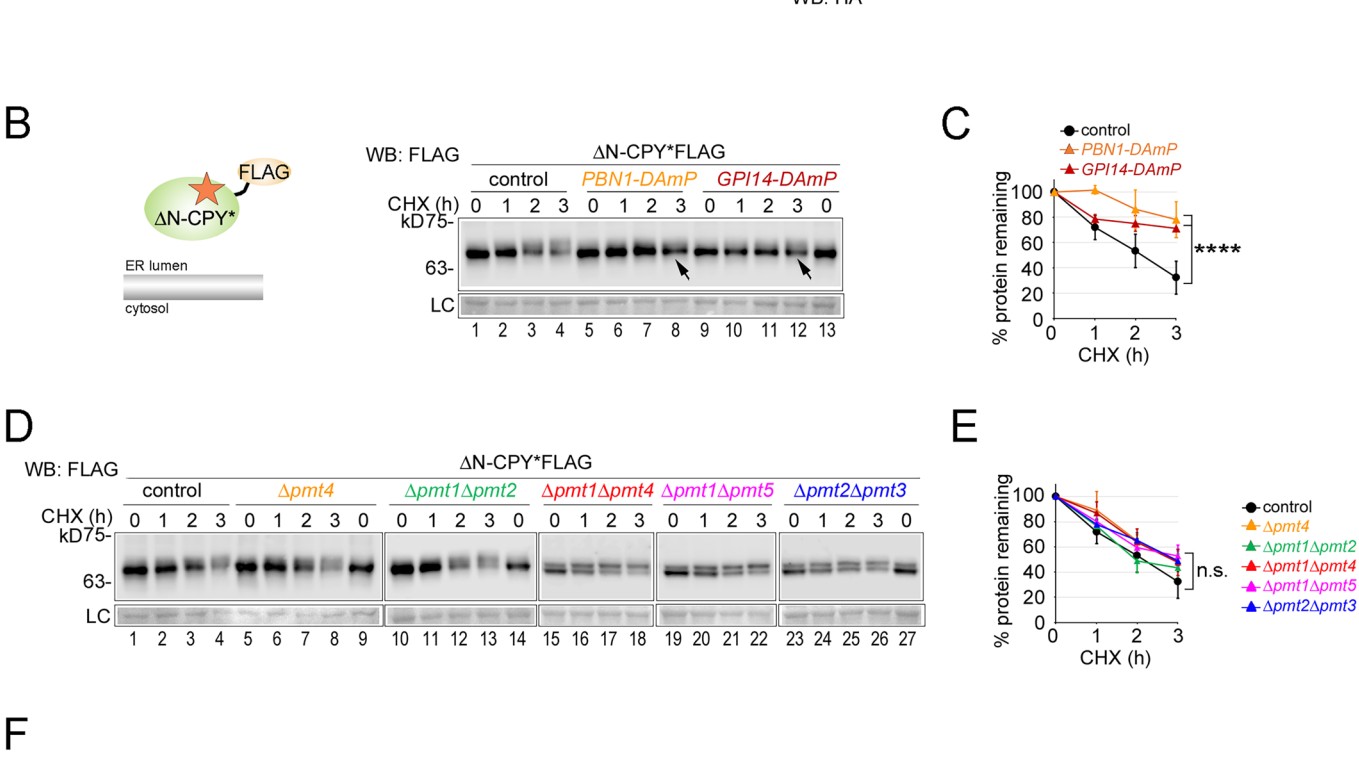

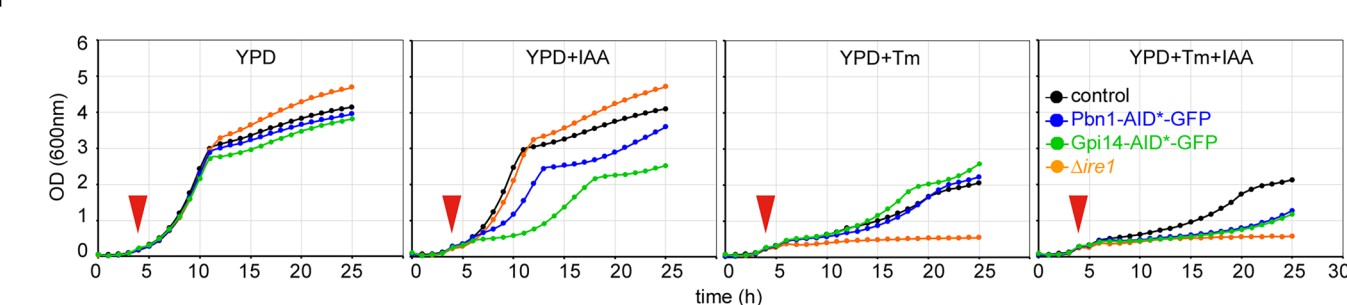

**Figure EV5. Functional characterization of Pbn1-Gpi14.**

(A) Control for concanavalin-A-mediated pulldown. Cells expressing plasmid-borne CFTR-ΔF208-HA (graphical depiction) were lysed, and the lysate was incubated with concanavalin-A-coupled beads (IP conA) for 3 h, followed by SDS-PAGE and WB analysis with antibodies against HA. CFTR-ΔF208-HA is not glycosylated in yeast, and its failure to be precipitated with conA served as a specificity control for the assay and validated the results obtained with ΔNg-CPY* (Fig. 5G). (B, C) Control cells and cells containing the *PBN1-DAmP* or *GPI14-DAmP* alleles were used to express plasmid-borne ΔNg-CPY*FLAG (graphical depiction) for CHX shut-off experiments, followed by SDS-PAGE and WB analysis. The arrows indicate an accumulation of ΔNg-CPY*FLAG with lower MW compared to the control. (D, E) Experiments using PMT deletion mutants. Yeast expresses six verified PMTs, with the most well-characterized being the conserved Pmt1, Pmt2, and Pmt4 (Lommel and Strahl, 2009). Control cells and cells with the indicated single and double deletions of PMT genes were used to express plasmid-borne ΔNg-CPY*FLAG for CHX shut-off experiments. Neither in the *Δpmt4* mutant, which lacks the homodimeric Pmt4 complex, nor in the *Δpmt1Δpmt2* double mutant, which lacks the heterodimeric Pmt1/2 complex, did we detect alterations in O-mannosylation of ΔNg-CPY* (lanes 1–14). The same results were obtained with those double mutants that simultaneously prevented the formation of the canonical and a cross-combinatorial Pmt1/2 complex (lanes 15–27). These results combined suggest that canonical PMTs do not participate in the O-mannosylation and degradation of ΔNg-CPY*. (F) Growth assays in the absence and presence of ER stress after acute depletion of either Pbn1 or Gpi14. The indicated strains were grown in rich medium (YPD) in the absence or presence of 1 µg/ml tunicamycin (Tm) and 500 µM auxin (indole-3-acetic acid (IAA)) at 30 °C. Drugs together with solvents or solvents alone were applied 4 h after starting the experiment by resetting cellular density (indicated by red arrowheads). Growth was determined by automated measurement of absorbance at 600 nm for individual cell cultures over a period of 25 h. The values are the mean of two independent measurements. Data information: Number of experiments (*n*) signifies biological replicates. Error bars in graphs represent the standard deviation from the mean. Statistical significance for all experiments involving degradation rates was calculated using two-way ANOVA (alpha = 0.05) with Šidák's correction for multiple comparisons, obtaining the following *p* values. (C): (*PBN1-DAmP*: $n = 4$, $p = 0.9 \times 10^{-4}$; *GPI14-DAmP*: $n = 4$, $p = 0.9 \times 10^{-4}$), (E): (*Δpmt4*: $n = 2$, $p = 0.6803$; *Δpmt1Δpmt2*: $n = 4$, $p = 0.9260$; *Δpmt1Δpmt4*: $n = 4$, $p = 0.2865$; *Δpmt1Δpmt5*: $n = 2$, $p = 0.8328$; *Δpmt2Δpmt3*: $n = 2$, $p = 0.5133$). n.s. = not significant; ****$p < 0.0001$.

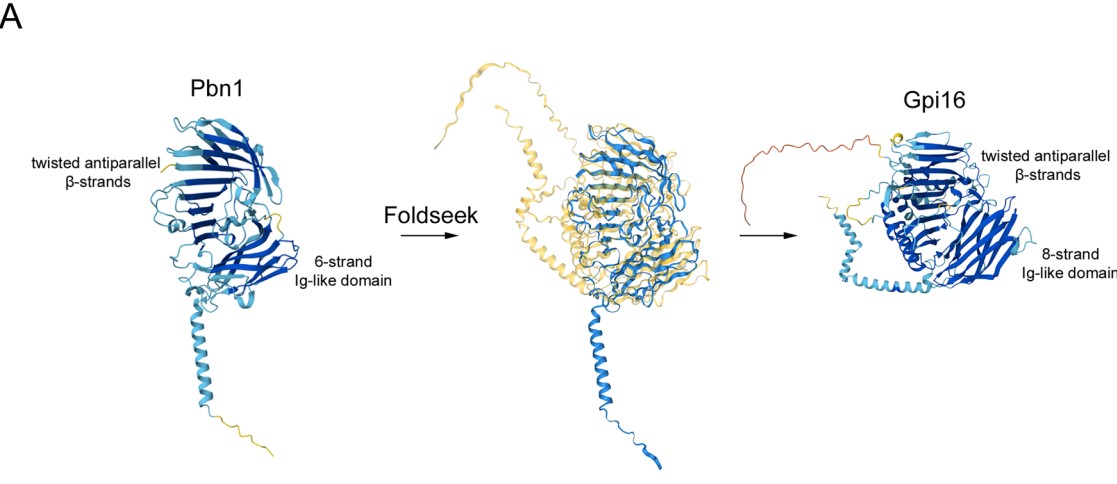

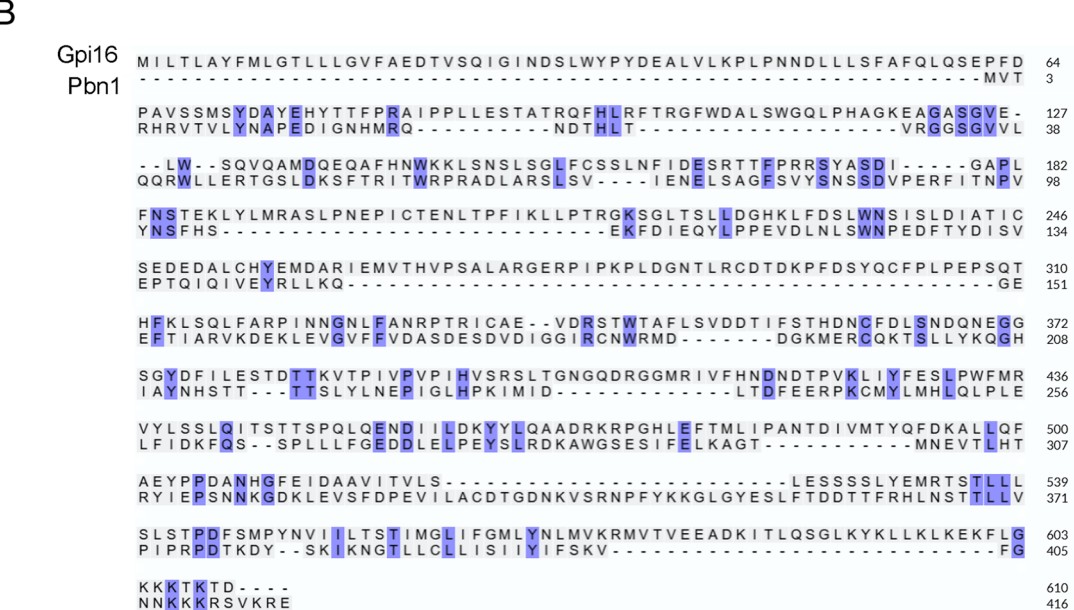

**Figure EV6. Search for proteins with similar folds to Pbn1.**

(A) Identification of Gpi16 as a protein structurally similar to Pbn1. Foldseek (van Kempen et al, 2024) was employed to identify structural similarities across proteins from all kingdoms using the Pbn1 Alphafold3 structure as bait (left). The essential component of the pentameric GPI-transamidase complex, Gpi16, emerged as the sole hit with a high degree of similarity. This is illustrated by the overlay (yellow) in the center. The Alphafold3 structure of Gpi16 is displayed on the right. (B) Low sequence identity between Pbn1 and Gpi16. Protein alignment analysis revealed a low overall sequence identity (18%) between Pbn1 and Gpi16. Conserved amino acids are highlighted in purple.

