## [Peer Review File · The EMBO Journal]

O-mannosylation of misfolded proteins promotes ERAD

Leticia Lemus, Hadar Meyer, Ana Rodriguez-Rosado, Maya Schuldiner, and Veit Goder *Corresponding*

author(s): Veit Goder (vgoder@us.es) , Leticia Lemus (llemus@us.es)

Review Timeline:

Submission Date:	25th Mar 25
Editorial Decision:	9th May 25
Revision Received:	16th Aug 25
Editorial Decision:	16th Oct 25
Revision Received:	19th Oct 25
Accepted:	4th Nov 25

Editor: William Teale

Transaction Report:

Dear Dr Goder,

Thank you again for the submission of your manuscript entitled " ER-associated degradation relying on protein O-mannosylation" (EMBOJ-2025-120895). I have now received three referees' reports, which are copied to the bottom of this message.

All referees agree that your conclusions are based on a technically well-performed series of experiments. They also state unambiguously that the manuscript is timely and the topic is important. However, the feedback was not unambiguously positive. The hypothesis you propose, that O-mannosylation by Pbn1 of misfolded serine-rich protein regions is sufficient for the targeting of substrates for ERAD, will need to be given stronger support if your manuscript is to be published in EMBO Journal.

All referees were concerned about the ability of the DAmP system to generate unambiguous data; here, there may be the possibility of extending your use of AID-mediated protein depletion (as referee #3 suggests). Additionally, Referees #2 and #3 mentioned that your work does not unambiguously show the involvement of SRR-O-mannosylation, and I think Referee #1 raises a valid point about indirect effects on the ERAD machinery itself. If you judge that such experiments are not feasible within a reasonable time-frame, it may be in your best interests to submit the study elsewhere. However, I am prepared formally to invite you to submit a revised version of the manuscript, and suggest we meet via Zoom (once you and you co-authors have had a chance to digest the reports) to discuss potential ways forward at EMBO Journal. Our usual revision time of three months is used as a guideline, not a deadline; manuscripts frequently take a little longer to revise.

I should add that it is The EMBO Journal policy to allow only a single major round of revision and that it is therefore important to resolve these concerns at this stage. Please contact me if you have any questions, need further input on the referee comments or if you anticipate any problems in addressing any of their points. Please, follow the instructions below when preparing your manuscript for resubmission.

I would also like to point out that as a matter of policy, competing manuscripts published during this period will not be taken into consideration in our assessment of the novelty presented by your study ("scooping" protection). We have extended this 'scooping protection policy' beyond the usual 3 month revision timeline to cover the period required for a full revision to address the essential experimental issues. Please contact me if you see a paper with related content published elsewhere to discuss the appropriate course of action.

Again, please contact me at any time during revision if you need any help or have further questions.

Thank you very much again for the opportunity to consider your work for publication. I look forward to your revision.

Best regards,

William Teale

William Teale, Ph.D.
Editor
The EMBO Journal

When submitting your revised manuscript, please carefully review the instructions below and include the following items:

- 1) a .docx formatted version of the manuscript text (including legends for main figures, EV figures and tables). Please make sure that the changes are highlighted to be clearly visible.
- 2) individual production quality figure files as .eps, .tif, .jpg (one file per figure).
- 3) a .docx formatted letter INCLUDING the reviewers' reports and your detailed point-by-point response to their comments. As part of the EMBO Press transparent editorial process, the point-by-point response is part of the Review Process File (RPF), which will be published alongside your paper.
- 4) a complete author checklist, which you can download from our author guidelines ([https://wol-prod-cdn.literatumonline.com/pb-assets/embo-site/Author Checklist%20-%20EMBO%20J-1561436015657.xlsx](https://wol-prod-cdn.literatumonline.com/pb-assets/embo-site/Author%20Checklist%20-%20EMBO%20J-1561436015657.xlsx)). Please insert information in the checklist that is

also reflected in the manuscript. The completed author checklist will also be part of the RPF.

6) We require a 'Data Availability' section after the Materials and Methods. Before submitting your revision, primary datasets produced in this study need to be deposited in an appropriate public database, and the accession numbers and database listed under 'Data Availability'. Please remember to provide a reviewer password if the datasets are not yet public (see <https://www.embopress.org/page/journal/14602075/authorguide#datadeposition>). If no data deposition in external databases is needed for this paper, please then state in this section: This study includes no data deposited in external repositories. Note that the Data Availability Section is restricted to new primary data that are part of this study.

Note - All links should resolve to a page where the data can be accessed.

8) For data quantification: please specify the name of the statistical test used to generate error bars and P values, the number (n) of independent experiments (specify technical or biological replicates) underlying each data point and the test used to calculate p-values in each figure legend. The figure legends should contain a basic description of n, P and the test applied. Graphs must include a description of the bars and the error bars (s.d., s.e.m.).

9) We would also encourage you to include the source data for figure panels that show essential data. Numerical data can be provided as individual .xls or .csv files (including a tab describing the data). For 'blots' or microscopy, uncropped images should be submitted (using a zip archive or a single pdf per main figure if multiple images need to be supplied for one panel). Additional information on source data and instruction on how to label the files are available at .

10) We replaced Supplementary Information with Expanded View (EV) Figures and Tables that are collapsible/expandable online (see examples in <https://www.embopress.org/doi/10.15252/embj.201695874>). A maximum of 5 EV Figures can be typeset. EV Figures should be cited as 'Figure EV1, Figure EV2" etc. in the text and their respective legends should be included in the main text after the legends of regular figures.

12) Our journal encourages inclusion of *data citations in the reference list* to directly cite datasets that were re-used and obtained from public databases. Data citations in the article text are distinct from normal bibliographical citations and should directly link to the database records from which the data can be accessed. In the main text, data citations are formatted as follows: "Data ref: Smith et al, 2001" or "Data ref: NCBI Sequence Read Archive PRJNA342805, 2017". In the Reference list, data citations must be labeled with "[DATASET]". A data reference must provide the database name, accession number/identifiers and a resolvable link to the landing page from which the data can be accessed at the end of the reference. Further instructions are available at .

13) In order to increase the reproducibility and reach of your work, The EMBO Journal includes a table of reagents that were used in the study. Please provide this along with your revisions

Further instructions for preparing your revised manuscript:

- a point-by-point response to the referees' comments, with a detailed description of the changes made (as a word file).
- a word file of the manuscript text
- individual production quality figure files (one file per figure)
- a complete author checklist, which you can download from our author guidelines (<https://www.embopress.org/page/journal/14602075/authorguide>).
- Expanded View files (replacing Supplementary Information)

- We ask authors to provide source data for the main manuscript figures. You will receive a separate email with instructions for providing source data with your revised manuscript, including how to upload and organize the files.

The revision must be submitted online within 90 days; please click on the link below to submit the revision online before 7th Aug 2025.

Referee #1:

This is an interesting manuscript reporting that the mannosyltransferase transferase complex Pbn1-Gpi14 influences ERAD in *S. cerevisiae*. However, I have some concerns, especially on the evidence that this is a direct effect of O-mannosylation of the misfolded protein substrates.

Major concerns:

1) The title, abstract and introduction are confusing and even misleading concerning the fact that this is a study in *S. cerevisiae*. Given the important differences in glycosylation and ERAD between yeast and higher eukaryotes, that should be clearly stated. The title should indicate that the observation is in yeast, or *S.cerevisiae*. The Introduction should have a paragraph explaining the differences in O-glycosylation between *S. cerevisiae* and higher eukaryotes. "Protein O-glycosylation, also known as O-mannosylation, is a second type of glycosylation..." as written in the Introduction is wrong for higher eukaryotes, where O-glycosylation starts mostly in the Golgi, by addition of O-GlcNAc. O-mannosylation in the ER is a minor pathway, restricted to few proteins in mammals.

2) It should be explained or speculated why PBN1 was the only gene identified in both screens (Fig. 1E).

3) Is it possible that Pbn1-Gpi14 manipulation might be influencing ERAD indirectly, by affecting GPI anchor synthesis which in turn could affect GPI anchoring of ERAD-modulating proteins? The authors show that Pbn1 does not indirectly affect PMT activity and therefore protein O-mannosylation (suppl. Fig. S2H). However, they do not rule out that an effect on GPI anchoring could affect directly ERAD component function

Minor concerns:

4) The effect of PBN2-DAmP on CPY* TMD seems very small and non-significant (Fig. 3F,G).

5) Why does deletion of any of the ERAD complexes reduced the degradation rate (Figs. 3J and K) (line 241-242)?

6) In the experiment in Δ emp24 cells, could it be that if exit to vacuoles is blocked, the degradation could switch to ERAD (suppl. Figs. S3C and D) (lines 242-243)?

7) Could it be that CPY* is also O-mannosylated to a small extent, but this is masked in the gels by the shifts by N-glycosylation?

8) Fig. 5I and 5K. It is difficult to see in the gels that the depletions result in an enrichment of the non-mannosylated form of Δ N-CPY* and not of the mannosylated form.

Referee #2:

Recognition of proteins for degradation from within the ER is an incompletely understood process. Classically, N-linked glycosylation sites are known to serve as a signal for degradation. However, in many cases, proteins are still degraded without having apparent N-linked glycans. Understanding the mechanism of substrate recognition for non-glycosylated proteins is still a work in progress. Recent work from other groups reported primary sequences that can serve as signals for degradation from the ER but Lemus et al have continued to study other mechanisms that could be in use.

In the manuscript by Lemus et al, the authors propose a role for o-linked mannosylation in recognition of substrates for degradation. They report two screens focused on turnover of a model substrate Gas1 that give mostly different results, but provide them with one overlapping hit named Pbn1. They suggest that Pbn1 shares homology with PIG-X and has some effect on the turnover of a misfolded GPI-anchored protein. The authors propose the SRR region from GPI1 is o-mannosylated and transfer this region (including the TM) to a known misfolded protein, and claim this fusion protein requires Pbn1 for degradation.

The idea that O-mannosylation is a fail-safe mechanism for directing ERAD, in the absence of N-glycosylation is intriguing. Unfortunately, the results are quite far from demonstrating this, or rigorously testing this idea. They fail to test their own model, which seems like pretty straightforward experiments that are necessary to publish. The experiments would be pretty straightforward, but the results are not guaranteed to support the model. The authors fail to provide enough insight for the conclusions to be well-justified.

Major concerns

- The effects observed in all cases, from either DAmP depletion or Gal1 overexpression are quite small. It is unclear whether this is actually a mechanism at use in cells.
- It seems like the fact that Pbn1 could have o-mannosylation activity towards a protein would need to be demonstrated. I understand this is not necessarily straightforward, but there is no evidence that this protein has this activity. This would really help establish that these modifications are o-mannosylation and catalyzed by Pbn1, which is central to this manuscript.
- The transfer of the SRR, without the associated TMD, seems like a way to establish this role for o-mannosylation. Additionally, because the SRR is attached to a known misfolded protein already, I don't know what we are really learning. To establish this role more directly, take the SRR, put it on different proteins (stable, unstable, soluble, integral membrane proteins) and show that this is sufficient to induce degradation.
- The experiments with the *pmt1Δpmt2Δ* strain do not really fit with the authors' model. The fact that removing these proteins speeds everything up, would seemingly contradict the model of the mechanic as a fail-safe. I understand that the substrates they tested might not fit this model, but then which proteins would? Having at least an example of this would be important to make this claim.
- All conclusions are based on experiments with a single model GPI-anchored misfolded substrate or a TM version of a second protein, CPY. The authors should show this with additional protein target(s).
- Figure 3 inexplicably uses the TMD in addition to the SRR, AFTER the authors show the TMD changes how Gas1* behaves (Figure 2G).
- The apparent partial requirement for Asi1, Doa10 and Hrd1 should be addressed (Figure 3J). This doesn't fit the conventional understanding of ERAD pathways and need to be reconciled, or removed. Perhaps they have multiple topologies of their substrate that could explain this result?
- Figure 4 does nothing to advance the conclusions of this manuscript and should just be removed. If the authors want to claim an association, in vivo imaging is completely insufficient to do so.
- In Figure 5G, the upper band proposed to be mannosylated CPY* is much more prominent than in any other experiment (5E,H,I,K+supplement). How do the authors know that this modification is not happening post-lysis?
- In the pulldown, using CFTR as a control protein is probably not appropriate, since it is a mammalian protein and a polytopic membrane protein.
- Figure 5A seems like it should exist in the supplement rather than the main text. The conclusions drawn are definitely not solid and can be explained by synthetic genetics effects.

Minor concerns

- I am not sure what the point of Figure 2A is. The homology between PIG-X and Pbn1 was previously reported.
- Proteins in the PIG pathways have been reported to be regulated by ERAD, so is it possible that these phenotypes are indirect?
- Somewhere, we need to see the overexpression of the Pbn1 protein that is being expressed from the Gal4 promoter. It is essential to know how this relates to the endogenous levels.
- In Figure 5H, the claim of 3 bands for lane 4 is very unconvincing. This seems to be quite simple to show in a more clear manner, simply by changing the percentage acrylamide and loading more material.
- The ΔN construct nomenclature is confusing for the CPY* missing n-linked glycosylation sites. I would suggest either coming up with another term or using the previous nomenclature for that construct.

Other suggestions

- Standard nomenclature in budding yeast is to have the Δ after the gene name (pmt1 Δ rather than Δ pmt1)

Referee #3:

Protein O-mannosylation is a widespread post-translational modification of secretory pathway proteins. Like protein N-glycosylation, it has roles in protein quality control in the endoplasmic reticulum but these roles are poorly defined. In the present study, Lemur and colleagues uncover a role of O-mannosylation in directing misfolded proteins towards ER-associated degradation (ERAD). Specifically, they use complementary genetic screens and subsequent protein degradation assays to show that Pbn1 and Gpi14, which are homologs of the human PIG-X and PIG-M proteins and have established roles in the biogenesis of GPI-anchored proteins (GPI-APs), together promote the O-mannosylation of misfolded proteins. In contrast to O-mannosylation by the Pmt1/2 mannosyl transferases, O-mannosylation by Pbn1-Gpi14 does not promote ER export but ERAD. This mechanism may be important for the degradation of both, proteins that are N-glycosylated and proteins that lack N-glycans. Thus, this study identifies new molecular machinery for ERAD, elaborates on the functions of O-mannosylation and puts it into the context of other mechanisms that serve the recognition and elimination of damaged proteins. Therefore, this study is a significant advance towards understanding protein quality control in the ER.

The experiments in the paper are carefully done and, for the most part, conclusive. The paper is well written and easy to follow. I have two suggestions for further improving the quality of the data (see major points below). Putting these suggestions into practice does not require any technical innovation, should be doable within a reasonable time frame and would substantially improve the manuscript. Besides these two major points, I only have a number of minor points (see below), all of which can be addressed editorially (unless the authors choose to take up the suggestion in minor point 14). If these points are addressed, I believe that the manuscript deserves publication in the EMBO Journal.

Major points

1. The authors did not detect a change in the degradation rate of Gas1**TMD* when Pbn1 function was impaired by means of the PBN1-DAmP allele (Figure 2G, H). On the other hand, PBN1 overexpression accelerated Gas1**TMD* degradation (Figure 2I, J). It is difficult to decide whether these observations indicate a direct role of Pbn1 in ERAD of non-GPI-APs. If Pbn1 had such a role, the experiments may have failed to reveal it because the reduction of Pbn1 levels through the DAmP allele was not strong enough or because cells had adapted to it. I recommend to attempt to clarify this point by acutely depleting Pbn1 with the AID system (as in Figure 5) and looking at Gas1**TMD* degradation again. Similarly, I suggest to improve consistency across experiments by testing HA-Gas1* and HA-Gas1wt degradation after auxin-induced Pbn1 depletion. Of the two systems used for Pbn1 depletion (DamP allele and AID system), the AID is by far superior because of the acute rather than chronic disruption of Pbn1 function it affords. Hence, the paper would greatly benefit if this system were used for all key experiments.
2. To explore the role of O-mannosylation in ERAD, the authors compared the degradation of soluble CPY*, which is not O-mannosylated, and CPY**TMD*, which is O-mannosylated due to a serine-rich region. Pbn1 is then shown to have a role in ERAD of CPY**TMD* but not CPY* (Figure 3). However, CPY**TMD* differs from CPY* in two regards: the presence of the serine-rich region and also anchoring in the ER membrane, and it is not clear which factor is responsible for the different susceptibility to Pbn1-mediated ERAD. For a clean comparison, it is necessary to isolate the possible role of the serine-rich region, either by investigating a version of CPY**TMD* that does not possess the serine-rich region or by testing a version of soluble CPY* to which the serine-rich region has been added.

Minor points

1. Abstract, line 53: "... through protein O-mannosylation is an essential component of ER quality control." The ERAD defects upon loss of Pbn1-Gpi14 function revealed in this study are rather subtle, so I think the word "essential" is too strong.
2. Introduction, line 65: "... retrotranslocation (...) followed by their ubiquitination and ..." This phrasing implies that proteins are retrotranslocated and only then ubiquitinated. Has this been established? To my understanding the order of events is not clear and ubiquitination may also occur during or even prior to retrotranslocation.
3. Introduction, line 93: "Drug-induced inhibition of protein N-glycosylation has been found to increase global protein O-mannosylation and reduce protein aggregation." As phrased, the sentence states that loss of N-glycosylation leads to less protein aggregation. Is this what the authors mean? Or do they mean that loss of N-glycosylation increases O-mannosylation and that O-mannosylation then helps to reduce protein aggregation?
4. Introduction, line 107: "... while a remaining pool ..." This should read 'the remaining pool'.
5. Results, line 181: "By combining genetics with enzyme treatment of cell lysates, we confirmed ..." I think it would help the reader to spell out that "enzyme treatment" here means removal of glycans with PNGase F.
6. Results, line 205: "In combination with the observed changes (...), it could suggest ..." I do not understand what 'it' refers to.
7. Results, line 263 and elsewhere: The tagged Gpi14 protein is here referred to as Gpi14-tdimer but as Gpi14-tdimer2 elsewhere, for example in the legend of Figure 4. Please edit the manuscript to be consistent throughout.
8. Results, line 309: The authors suggest that "(...) the observed connection of Pbn1 to protein O-mannosylation and ERAD includes Gpi14 and is important for a variety of misfolded proteins, including those that lack N-glycans and do not contain an SRR." Alternatively, Pbn1 and Gpi14 depletion may similarly impacted cell sensitivity to tunicamycin (Figure 5A) because GPI-AP biogenesis is impaired. Disrupted GPI-AP biogenesis could increase ER stress and thus further slow cell growth in the presence of tunicamycin. I therefore believe that the data are consistent with epistasis between Pbn1 and Gpi14 but that they do not allow conclusions about which of their shared functions (GPI-AP biogenesis or ERAD) are responsible for the observed growth phenotypes.
9. Results, line 320: To help the reader, I suggest editing the text to "..., albeit less efficiently than CPY* (Figs. 5E and F, compare with Figs. 3A and B)."
10. Results, line 334: I stumbled over the term "enrichment" because it was not clear to me relative to what the non-mannosylated form of ΔN -CPY* was enriched. Instead of enrichment, the term 'accumulation' might be simpler.
11. Results, line 345: "Additionally, ΔN -CPY* degradation was stabilized and ...". Shouldn't "degradation" be deleted here?
12. Results, line 351: "... and are consistent with the observed sensitivity of cells to tunicamycin upon depletion of either Pgn1 or Gpi14 (Fig. 5A)." Since the data in Fig. 5A may be open to alternative interpretation (see minor point 8), it may be better to delete this part of the sentence and not muddy the conclusion drawn from Figure 5I - L.
13. Discussion, line 371: The authors state that their data suggest a role of O-mannosylation in both ERAD of N-glycosylated proteins and ERAD of proteins that lack other ERAD cues. The next paragraph begins with "Our findings related to SRRs support the first scenario", which makes it sound as if lines 371-373 had stated two mutually exclusive possibilities. Please edit for clarity, for instance by starting line 374 with something like: Our data provide evidence for both scenarios. First, our findings related to SRRs ...
14. Discussion, line 393: The authors make the reasonable and testable suggestion that Yos9 may function in the recognition of ERAD cues based on O-mannosylation. It is a pity that they did not test this idea directly using ΔN -CPY*, and the reader is left wondering why this relatively simple experiment wasn't done. I therefore encourage the author to consider investigating the possible role of Yos9 already in this paper.
15. Discussion, line 396: see minor point 13.
16. Discussion, starting at line 406: I strongly agree with this paragraph, which provides a rationale to extend the use of the AID system to the experiments shown in Figure 2 (also see major point 1).
17. Discussion, line 449: The discussion ends rather abruptly, and a brief summary paragraph would be nice.
18. Discussion: The data convincingly support a role for Pbn1/Gpi14-mediated O-mannosylation in ERAD, and the authors

appear to favour the idea that O-glycosylation acts as ERAD "cues". However, is it possible that this O-mannosylation acts to end futile folding cycles as proposed by Xu et al., 2013? Unless this idea can be discarded for some obvious reason, the authors should bring it up as an alternative model to lectin-mediated recognition of O-mannosylation-based degrons.

19. Overall, the paper is quite concise overall with just five figures. The authors could therefore consider integrating some of supplemental data into main figures to make them more visible. This, however, is just an idea the authors can feel free to ignore.

Point-by-point response

REFEREE #1:

We would like to thank the Reviewer for their dedication to revising our manuscript and for the valuable comments that helped improve our work. In response to the comments, we have conducted additional experiments and revised the text accordingly. Below, we provide a point-by-point response to the original comments. We hope that the revised version of our manuscript will be received favorably by the Reviewer.

This is an interesting manuscript reporting that the mannosyltransferase transferase complex Pbn1-Gpi14 influences ERAD in *S. cerevisiae*. However, I have some concerns, especially on the evidence that this is a direct effect of O-mannosylation of the misfolded protein substrates.

Major concerns:

1) The title, abstract and introduction are confusing and even misleading concerning the fact that this is a study in *S. cerevisiae*. Given the important differences in glycosylation and ERAD between yeast and higher eukaryotes, that should be clearly stated. The title should indicate that the observation is in yeast, or *S.cerevisiae*. The Introduction should have a paragraph explaining the differences in O-glycosylation between *S. cerevisiae* and higher eukaryotes. "Protein O-glycosylation, also known as O-mannosylation, is a second type of glycosylation..." as written in the Introduction is wrong for higher eukaryotes, where O-glycosylation starts mostly in the Golgi, by addition of O-GlcNAc. O-mannosylation in the ER is a minor pathway, restricted to few proteins in mammals.

We agree with the reviewer that the term "O-glycosylation" may be confusing when used synonymously with O-mannosylation, as the sentence of reference may have implied. This distinction is important to avoid misinterpretation with other types of O-glycosylation that occur predominantly in the Golgi and are catalyzed by distinct enzymes, as the reviewer pointed out. Accordingly, we have revised the text of our manuscript to consistently use the term "protein O-mannosylation" and have removed the term "O-glycosylation" from the critical passage (lines 75-81). In addition, we have clarified that ER-localized protein O-mannosyltransferases linked to this study are conserved between yeast and higher eukaryotes, while others are specific to higher eukaryotes. We have also included an additional reference to specific protein O-mannosyltransferases in mammalian cells to support this point.

2) It should be explained or speculated why PBN1 was the only gene identified in both screens (Fig. 1E).

PBN1 emerged as the sole overlapping hit in both screens. This is essentially due to the thresholding applied to the data obtained with the second screen. We annotated hits when fluorescence decreased by more than 20% compared to the medium value. These criteria for hit selection are explained in the legend to Table EV1. We have now included additional text in the Result section to better explain this fact to the readers. (lines 153-157)

3) Is it possible that Pbn1-Gpi14 manipulation might be influencing ERAD indirectly, by affecting GPI anchor synthesis which in turn could affect GPI anchoring of ERAD-modulating proteins? The authors show that Pbn1 does not indirectly affect PMT activity and therefore protein O-mannosylation (suppl. Fig. S2H). However, they do not rule out that an effect on GPI anchoring could affect directly ERAD component function.

We have emphasized this control. Using both the non-N-glycosylated Δ Ng-CPY* and the canonical N-glycosylated CPY* as model substrates, we have shown that Pbn1 depletion affected the degradation of only Δ Ng-CPY*, showing that ERAD was not generally impaired at this condition (these data are shown in Fig. 5H,I and 5L,M in the revised figure order). (lines 348-352)

To provide additional evidence beyond the conditions involving acute Pbn1 depletion, we have now generated and characterized a series of new Pbn1 mutants. While most mutants were either nonviable or showed no discernible phenotype, the Pbn1-TMD:LIV mutant, which contains an altered transmembrane domain (TMD), exhibited specific ERAD-related phenotypes. Similar to the results observed with acutely depleted Pbn1, Pbn1-TMD:LIV selectively inhibited the degradation of Δ Ng-CPY*, while the degradation of CPY* remained unaffected (compare Fig. 6I, J with Fig. 6K, L). (lines 394-398)

With the inclusion of these additional experiments, we now present a more comprehensive argument that the inhibition of Δ Ng-CPY* degradation is not due to a general impairment of ERAD function resulting from potential GPI anchoring defects under the same conditions.

Minor concerns:

4) The effect of PBN2-DAmP on CPY*TMD seems very small and non-significant (Fig. 3F,G).

Although changes in degradation rates might appear small (or large), the significance of each observation was determined using statistical tests and all numerical data (including p-values) are now additionally appended to the individual Figure legends. The observations made with the construct CPY*SRR_TMD (and with the new constructs CPY*SRR and CPYwtSRR) were used to formulate the hypothesis that in absence of N-glycans, the effects based on Pbn1-dependent protein O-mannosylation might be more prominent. Indeed, degradation defects become more drastic when using the non-N-glycosylated model protein Δ Ng-CPY* (Fig. 5 and 6). We have now introduced additional text to explain this connection better. (lines 319-327)

5) Why does deletion of any of the ERAD complexes reduced the degradation rate (Figs. 3J and K) (line 241-242)?

This is an interesting but not overly surprising observation and might in part also be linked to the overexpression of the substrate. However, redundancy between different ERAD systems for misfolded substrates in yeast has been reported previously, for instance: (1) Misfolded Yor1 uses both Doa10 and Hrd1 complex (S. Pagant et al., 2007, PMID: 17615300), (2) Sec61-2 is an ERAD substrate for both the Asi-complex and the Hrd1-complex (O. Foresti et al., 2014, PMID: 25236469), or (3) misfolded amino acid permeases (AAPs) are degraded by both the Doa10 and the Hrd1 complex (J. Kota et al., 2007, PMID: 17325204). Because of this, and due to space constraints, we have refrained from providing a more detailed explanation of the observed redundancies at this stage.

6) In the experiment in $\Delta emp24$ cells, could it be that if exit to vacuoles is blocked, the degradation could switch to ERAD (suppl. Figs. S3C and D) (lines 242-243)?

In general, dynamic rerouting of substrates between ERAD and the vacuole in an *in vivo* environment has been observed and reported (Goder and Melero, 2011, PMID: 21147851; Sikorska et al., 2016, PMID: 27325793; Molinari, 2021, PMID: 33765438). The exit to the vacuole is not generally blocked in $\Delta emp24$ cells - it mostly affects the ER export of GPI-APs - and CPY*SRR_TMD was therefore not expected to be severely affected. We used this strain to rule out that protein stabilization upon deletion of ERAD machinery components could be indirect through altered ER homeostasis rather than due to a direct ERAD defect. Since $\Delta emp24$ elicits a UPR response (Belden and Barlow, 2001, PMID: 11294899), we thought this to be a good control to address these potential indirect effects. We have now introduced additional text to explain this better. (lines 255-261)

7) Could it be that CPY* is also O-mannosylated to a small extent, but this is masked in the gels by the shifts by N-glycosylation?

N-glycosylation does not appear to mask *per se* the additional post-translational increase in MW caused by O-mannosylation, exemplified with Gas1*, CPY*SRR_TMD and CPY*SRR, all of which are also N-glycosylated. It therefore appears that the only limitation for detection using our assays would be the modification with too few mannoses. That said, we do not rule out that CPY* is O-mannosylated to some extent even under normal cellular growth conditions. However, it would be minute, since we found CPY* to be degraded completely independent of Pbn1 function and since it does not show any conventionally detectable heterogeneity in MW-in contrast to ΔNg -CPY*.

8) Fig. 5I and 5K. It is difficult to see in the gels that the depletions result in an enrichment of the non-mannosylated form of ΔN -CPY* and not of the mannosylated form.

It is possible that in the miniaturized Figure (print size), the differences are less apparent. However, with minor enlargement, which would be easily achievable with the zoom function in the online version, the differences are clearly visible. If the Reviewer or the Editor feels that the Zoom function would be insufficient to detect the differences, we could prepare an enlarged part of the Figure to be shown in the EV material.

REFEREE #2

We sincerely thank the Reviewer for their thorough and insightful critique of our manuscript. Their constructive suggestions have significantly strengthened our work. We have fully addressed the Reviewer's valuable concerns regarding our conclusions in the revised manuscript and have extensively modified text passages, tuned down some of our conclusions, re-arranged figures and performed additional experiments. We believe the revisions have significantly improved the manuscript and hope it now meets the Reviewer's approval.

Recognition of proteins for degradation from within the ER is an incompletely understood process. Classically, N-linked glycosylation sites are known to serve as a signal for degradation. However, in many cases, proteins are still degraded without having apparent N-linked glycans. Understanding the mechanism of substrate recognition for non-glycosylated proteins is still a work in progress. Recent work from other groups reported primary sequences that can serve as signals for degradation from the ER but Lemus et al have continued to study other mechanisms that could be in use.

In the manuscript by Lemus et al, the authors propose a role for o-linked mannosylation in recognition of substrates for degradation. They report two screens focused on turnover of a model substrate Gas1 that give mostly different results, but provide them with one overlapping hit named Pbn1. They suggest that Pbn1 shares homology with PIG-X and has some effect on the turnover of a misfolded GPI-anchored protein. The authors propose the SRR region from GPI1 is o-mannosylated and transfer this region (including the TM) to a known misfolded protein, and claim this fusion protein requires Pbn1 for degradation.

The idea that O-mannosylation is a fail-safe mechanism for directing ERAD, in the absence of N-glycosylation is intriguing. Unfortunately, the results are quite far from demonstrating this, or rigorously testing this idea. They fail to test their own model, which seems like pretty straightforward experiments that are necessary to publish. The experiments would be pretty straightforward, but the results are not guaranteed to support the model. The authors fail to provide enough insight for the conclusions to be well-justified.

Major concerns

The effects observed in all cases, from either DAmP depletion or Gal1 overexpression are quite small. It is unclear whether this is actually a mechanism at use in cells.

We think that this statement relates to our initial experiments shown in Figures 2 and 3, where we used CHX shut-off experiments to validate our screen data (Figure 2) and to test our initial hypothesis for a potential function of Pbn1 in the context of protein O-mannosylation and ERAD together with the specifically generated chimeric construct

CPY*SRR_TMD (formerly CPY*TMD) alongside their controls (Figure 3). We agree that the observed differences in degradation rate with this reporter in DAmP cells or upon overexpression were small, yet they were significant ($p < 0.05$) (Fig3F-I). We later proposed that the simultaneous presence of N-glycans, which can form potent ERAD cues, might reduce the contribution of protein O-mannosylation to ERAD. After adding text, this is now explicitly explained (lines 319-327). Indeed, in absence of N-glycans, O-mannosylation contributes significantly to ERAD, exemplified by experiments performed with the construct Δ Ng-CPY* (Fig 5, EV5 and 6). These cases exhibit substantially stronger phenotypes and significantly more pronounced changes in degradation rate.

In addition, we have now better commented our conclusions drawn from each experiment and have refined our working hypotheses based on the experimental evidence presented throughout the manuscript. In the following we provide a short summary of the changes we have incorporated into the text: Based on our experiments shown in Figure 2 we formulated an initial hypothesis that Pbn1 affects the ERAD of GPI-APs, directly or indirectly (lines 175-177). However, this hypothesis was ruled out when Gas1*TMD, which lacks GPI anchor, was significantly affected upon Pbn1 overexpression, similar to Gas1*. This is now better explained in modified text passages: (lines 196-201). We have also changed text to better explain that based on these results linked to non-GPI-APs we have built a new working hypothesis that includes a function of Pbn1 in protein O-mannosylation (lines 220-226). Based on subsequent experiments using the constructs CPY*SRR_TMD (formerly CPY*TMD), CPY*SRR, and CPYwtSRR—all of which contain several N-glycans (strong ERAD cues) and an SRR—we have revised and expanded the text to clarify our rationale for next examining a misfolded protein lacking both N-glycans and an SRR, in order to more directly assess the role of protein O-mannosylation in ERAD (lines 319-330). Indeed, compared to the N-glycosylated CPY* variants, the non-N-glycosylated model protein Δ Ng-CPY* showed markedly stronger degradation defects in experiments using PBN1 or GPI14 DAmP alleles, under conditions of acute Pbn1 or Gpi14 depletion, or in the presence of a newly generated Pbn1 mutant (Fig. 5; new Fig. 6; lines 393–398).

It seems like the fact that Pbn1 could have o-mannsylation activity towards a protein would need to be demonstrated. I understand this is not necessarily straightforward, but there is no evidence that this protein has this activity. This would really help establish that these modifications are o-mannosylation and catalyzed by Pbn1, which is central to this manuscript.

As the reviewer pointed out, our experiments do not include *in vitro* data that directly address protein O-mannosylation activity for a receptor peptide.

Several lines of evidence from our experiments performed *in vivo* support the proposed bifunctional activity of Pbn1-Gpi14, including its role as a protein O-mannosyltransferase.

- 1) Our findings originate from an unbiased screen and have been confirmed with a variety of subsequent experiments. We present evidence that Pbn1 specifically affects O-mannosylation and degradation of various misfolded model proteins.

Addressing potential long-term adaptation with strains harboring the PBN1-DAmP allele, we have now included new experiments with acute depletion of Pbn1, using the AID*-degron, obtaining essentially the same results (Fig. 2O-U).

- 2) The ERAD pathway is not indirectly affected by Pbn1 depletion (either acutely or by expressing the PBN1-DAmP allele) since degradation of the canonical misfolded substrate CPY* remains unchanged under this condition (Fig. 3A,B; Fig5L,M). In addition, we have now performed and included new experiments with a novel Pbn1 mutant (Pbn1-TMD:LIV), which again specifically affects the degradation of Δ Ng-CPY* but not of CPY* (new Fig. 6I-L).
- 3) An unbiased phylogenetic analysis revealed that among members of the GT-C superfamily of mannosyltransferases, Gpi14 is closest related to PMTs (Fig. 4D).
- 4) In mammalian cells, a similar scenario is found where enzymes remotely related to POMTs (the orthologues of yeast PMTs) possess protein O-mannosyltransferases activity towards a subset of proteins inside the ER (Larsen ISB, et al., 2017, PMID: 28973932; Larsen ISB, et al., 2023, PMID: 37186866). This shows that a diverse population of related enzymes is capable of protein O-mannosylation.

The transfer of the SRR, without the associated TMD, seems like a way to establish this role for o-mannosylation. Additionally, because the SRR is attached to a known misfolded protein already, I don't know what we are really learning. To establish this role more directly, take the SRR, put it on different proteins (stable, unstable, soluble, integral membrane proteins) and show that this is sufficient to induce degradation.

We sincerely appreciate this insightful suggestion. In response, we have generated novel constructs lacking the TMD, including both misfolded SRR-containing CPY* and properly folded SRR-containing CPY to further address this important question. The experiments are shown in the new Fig. 3L-S. These data suggest that Pbn1 participates in the O-mannosylation and degradation of CPY*SRR, whereas CPYWTSRR is neither visibly O-mannosylated nor is its basal turnover affected by Pbn1 depletion. These data suggest that an SRR serves as a platform that can be modified by O-mannosylation when the protein is misfolded. We also introduced additional text to clarify that we consider an SRR not a degron *per se*, but that its modification can convert it into a degron (lines 426-428). This is also in agreement with the observations made originally with Gas1WT and Gas1* (Fig. 2K,L,T,U). Only the misfolded protein Gas1* was shown to be heavily modified by the Pmt1/2 complex with effects on ERQC and ERAD (Fig. EV2F and Hirayama et al., 2008, PMID: 18182384; Melero and Goder, 2011, PMID: 21147851), whereas Gas1WT was not modified by these PMTs (Hirayama et al., 2008, PMID: 18182384).

The experiments with the *pmt1Δpmt2Δ* strain do not really fit with the authors' model. The fact that removing these proteins speeds everything up, would seemingly contradict the model of the mechanic as a fail-safe. I understand that the substrates they tested might not fit this model, but then which proteins would? Having at least an example of this would be important to make this claim.

We have now modified text to unambiguously connect the phrase “fail-safe” to ERAD, not to general degradation, which would include vacuolar degradation (lines 419-423).

We agree that the phenotypes obtained with the deletion of PMTs or the depletion of Pbn1 appear contradictory at first sight. We have now extensively modified the paragraph in order to explain in more detail how the different phenotypes were obtained and why we think that PMTs and Pbn1 affect the substrate differently. We have previously shown that faster degradation of Gas1* variants after deletion of PMT1 and PMT2 is not only resulting in a loss of protein O-mannosylation but to a parallel occurring loss in ER retention, which causes efficient re-routing to the vacuole (Goder and Melero, 2011; PMID: 21147851). Thus, PMT1 and PMT2 combine protein O-mannosylation with ER retention for certain substrates. Since depletion of Pbn1 did also reduce protein O-mannosylation of several tested model substrates but did not produce faster degradation but rather stabilization, we reasoned that Pbn1 does not combine protein O-mannosylation with ER retention. We have then performed additional experiments using the established GFP cleavage assay to directly address this with the model substrate CPY*SRR_TMD. Indeed, we found that Pbn1 depletion did not increase the population of protein routed to the vacuole, unlike cells with PMT1 and PMT2 deleted, shown in Fig. EV3E,F. We have provided a more detailed explanation of these results and hope this will help prevent the emergence of contradictory conclusions (lines 271-282).

All conclusions are based on experiments with a single model GPI-anchored misfolded substrate or a TM version of a second protein, CPY. The authors should show this with additional protein target(s).

We believe the reviewer is referring to our conclusions related to the SRR, which we have also addressed above. However, the central conclusion of our manuscript—that protein O-mannosylation can promote ERAD—is primarily based on observations made with the misfolded model protein ΔNg-CPY*, which lacks both N-glycans and an SRR. These experiments with ΔNg-CPY* represent an independent line of evidence, unrelated to the SRR. In fact, the SRR- and N-glycan-containing model proteins were used in the context of follow-up experiments stemming from our initial screen, and the study was progressively refined to culminate in the analysis of ΔNg-CPY*.

With respect to the experiments linked to an SRR, we have now extended our analysis as the reviewer suggested and have performed additional experiments with newly generated constructs (new Fig. 3L-S). We have also added new text in our manuscript to explain these results in more detail (see also above).

Figure 3 inexplicably uses the TMD in addition to the SRR, AFTER the authors show the TMD changes how Gas1* behaves (Figure 2G).

In our initial attempt to test whether transferring an SRR-containing protein sequence would promote protein O-mannosylation and Pbn1 dependency, we used an SRR in combination with a TMD, for the ease of construction. Although we could observe O-mannosylation and Pbn1-dependency, we understand that our data did not warrant a conclusion for constructs that do not contain the TMD. As reported above, we have now addressed this point and have performed additional experiments with constructs that are devoid of a TMD (Fig. 3L-S).

The apparent partial requirement for Asi1, Doa10 and Hrd1 should be addressed (Figure 3J). This doesn't fit the conventional understanding of ERAD pathways and need to be reconciled, or removed. Perhaps they have multiple topologies of their substrate that could explain this result?

Regarding these results, what we have observed is a redundancy in degradation. This observation might in part be linked to the overexpression of the substrate but does not limit our conclusion that ERAD is involved in the degradation of the model protein. Redundancy between different ERAD systems for misfolded substrates in yeast has been reported previously, for instance: (1) Misfolded Yor1 uses both Doa10 and Hrd1 complex (S. Pagant et al., 2007; PMID: 17615300), (2) Sec61-2 is an ERAD substrate for both the Asi-complex and the Hrd1-complex (O. Foresti et al, 2014; PMID: 25236469), or (3) misfolded amino acid permeases (AAPs) are degraded by both the Doa10 and the Hrd1 complex (J. Kota et al., 2007; PMID: 17325204). Because of this, we have not explained the observed redundancies in more detail at this stage. We don't think that the redundancies could be explained by different substrate topologies because the model protein that we used for the experiment is N-glycosylated and mislocalization of the luminal domain to the cytosol (inverted topology) would lead to a non-N-glycosylated protein fraction, clearly distinguishable in MW from the N-linked protein. However, we do not see such a fraction. All protein seems to be exposed to the ER lumen.

Figure 4 does nothing to advance the conclusions of this manuscript and should just be removed. If the authors want to claim an association, *in vivo* imaging is completely insufficient to do so.

What we show is a co-localization of the components Pbn1 and Gpi14 *in vivo*, both are the key components in this study. We would prefer if this information was not removed. For instance, together with the subsequent experiments addressing a functional connection between these two proteins, we propose the existence of a protein complex. We agree with the reviewer that *in vivo* imaging is normally insufficient to prove any physical association. However, in this case we have made use of an unusual observation with genomic N-terminally GFP-tagged Pbn1, which forms cellular clusters, without compromising the essential protein function. Precisely the observation that Gpi14 co-localizes with Pbn1 under

these conditions, while remaining homogeneously ER-distributed in the absence of Pbn1 clusters, strongly supports their inclusion in the same cellular complex. In fact, it looks as if most cellular Pbn1 and Gpi14 are in the complex, since not much additional fluorescence is visible in the rest of the ER when clusters are formed.

We have now better explained the experiment and the conclusions. We have now also changed “association” with “co-localization”. At the same time, we mentioned that our observations support the notion that both components are part of a shared complex. (lines 295-298)

In Figure 5G, the upper band proposed to be mannosylated CPY* is much more prominent than in any other experiment (5E,H,I,K+supplement). How do the authors know that this modification is not happening post-lysis?

We consistently observe slightly different ratios of the mannosylated versus non-mannosylated Δ Ng-CPY* in all our experiments and these ratios also change throughout a CHX shut-off experiment, due to protein O-mannosylation and degradation occurring in parallel. The experiment shown in Fig. 5F (formerly Fig 5G) was based on a protocol that differed from most other types of experiments that are shown and involved heat treatment of cellular lysate and subsequent lengthy incubation at 37°C. This leads to a general loss in signal and increased background. However, we rule out post-lysis effects for the following reasons: First, before incubation with mannosidase, the lysate is incubated at 95°C in the presence of SDS, effectively inactivating all cellular enzymes. Second, for the CHX shut-off experiments with those proteins that show clearly visible protein fractions with higher MW, post-lysis (rather than cellular) O-mannosylation would imply that a constant fraction of total available substrate would be seen modified in the same experimental setup. This would mean that during a chase experiment, the ratio of mannosylated versus non-mannosylated Δ Ng-CPY* would remain constant over time. However, we do not observe a constant ratio in these experiments.

In the pulldown, using CFTR as a control protein is probably not appropriate, since it is a mammalian protein and a polytopic membrane protein.

We think that for the purpose of this experiment, it was merely important to have a non-N-linked control protein, to demonstrate that conA-Sepharose does not unspecifically bind non-glycosylated protein. In fact, the nature of CFTR, as a polytopic membrane protein, is also useful as a control to show that the experimental conditions we used did not lead to unwanted protein aggregation and unspecific precipitation, which is more likely to occur with polytopic membrane proteins, compared to a soluble protein like Δ Ng-CPY*.

Figure 5A seems like it should exist in the supplement rather than the main text. The conclusions drawn are definitely not solid and can be explained by synthetic genetics effects.

We agree with the reviewer that our data are merely consistent with a general role of Pbn1 and Gpi14 in maintaining ER homeostasis but not evidential. We have tuned down our

statement and have moved the figure to the Expanded View section (now Fig. EV5F).

Minor concerns

I am not sure what the point of Figure 2A is. The homology between PIG-X and Pbn1 was previously reported.

Only functional data between PIG-X and Pbn1 has been previously shown, but no structural comparison. We find this extremely insightful and illustrative, in particular with respect to the luminal domain(s). We have further elaborated on this for the generation of new Pbn1 mutants (new Fig.6).

Proteins in the PIG pathways have been reported to be regulated by ERAD, so is it possible that these phenotypes are indirect?

We agree that this is an important point to address. In addition to our previous experiments for addressing this point (reordered Fig. 5H-M), we have now included new data with a newly generated Pbn1 mutant (Fig. 6I-L). Both data sets show that only the ERAD of Δ Ng-CPY* but not of CPY* is inhibited in Pbn1-deficient cells, ruling out that potential interference with GPI metabolism under the same conditions would affect general ERAD function.

Somewhere, we need to see the overexpression of the Pbn1 protein that is being expressed from the Gal4 promoter. It is essential to know how this relates to the endogenous levels.

In general, the cellular expression level of Pbn1 is weak and the GAL1 promoter causes a high degree of Pbn1 overexpression. We have used differentially tagged versions of Pbn1 for expression testing. Expression of HA-tagged Pbn1 under the control of the GAL1 promoter in presence of galactose is easily detectable and is shown in Fig. EV2G. Using the same conditions and number of cells, we were unable to detect HA-tagged Pbn1 when expressed from the endogenous promoter, suggesting that the protein is highly expressed when driven by the GAL1 promoter. We show an additional experiment with GFP-tagged Pbn1 expressed from either the endogenous promoter or the moderate constitutive NOP1 promoter that allows direct comparison, shown in Fig. EV4A. Given that the GAL1 promoter is stronger in the presence of galactose than the constitutive NOP1 promoter, this experiment supports the conclusion that Pbn1 expression from the endogenous promoter can be several orders of magnitude lower than when driven by the GAL1 promoter.

In Figure 5H, the claim of 3 bands for lane 4 is very unconvincing. This seems to be quite simple to show in a more clear manner, simply by changing the percentage acrylamide and loading more material.

We have now repeated the experiment and have included new data. We can repeatedly precipitate a clearly visible additional protein fraction with higher MW in the Hrd1 deletion strain compared to the control strain (new Fig. 5G).

The Δ N construct nomenclature is confusing for the CPY* missing n-linked glycosylation sites. I would suggest either coming up with another term or using the previous nomenclature for that construct.

Previous work has used the term Δ g-CPY* for the non-glycosylated version of CPY*, referring to the non-N-linked protein. We felt that in the context of our study, the term Δ g-CPY* might be misleading since we find it to be O-mannosylated. Given these considerations, we changed it to Δ Ng-CPY* to account for the lack of N-glycosylation and maintaining the “core” of the original name.

Other suggestions

Standard nomenclature in budding yeast is to have the Δ after the gene name (pmt1 Δ rather than c)

We thank the reviewer for their observation and acknowledge that the nomenclature they reference is widely regarded as standard. However, given that variations remain in use across the field, including by our group, we believe the nomenclature adopted in our manuscript is appropriate and unlikely to cause confusion. With the reviewer’s consent, we would prefer to retain it, unless the editor advises otherwise.

REFEREE #3

We sincerely thank the Reviewer for their time, effort, and thoughtful evaluation of our manuscript. We are especially grateful for the constructive suggestions and detailed comments, which have greatly contributed to improving the quality of our work. We have carefully addressed each of the Reviewer's points, and our detailed responses are provided below.

Protein O-mannosylation is a widespread post-translational modification of secretory pathway proteins. Like protein N-glycosylation, it has roles in protein quality control in the endoplasmic reticulum but these roles are poorly defined. In the present study, Lemur and colleagues uncover a role of O-mannosylation in directing misfolded proteins towards ER-associated degradation (ERAD). Specifically, they use complementary genetic screens and subsequent protein degradation assays to show that Pbn1 and Gpi14, which are homologs of the human PIG-X and PIG-M proteins and have established roles in the biogenesis of GPI-anchored proteins (GPI-APs), together promote the O-mannosylation of misfolded proteins. In contrast to O-mannosylation by the Pmt1/2 mannosyl transferases, O-mannosylation by Pbn1-Gpi14 does not promote ER export but ERAD. This mechanism may be important for the degradation of both, proteins that are N-glycosylated and proteins that lack N-glycans. Thus, this study identifies new molecular machinery for ERAD, elaborates on the functions of O-mannosylation and puts it into the context of other mechanisms that serve the recognition and elimination of damaged proteins. Therefore, this study is a significant advance towards understanding protein quality control in the ER.

The experiments in the paper are carefully done and, for the most part, conclusive. The paper is well written and easy to follow. I have two suggestions for further improving the quality of the data (see major points below). Putting these suggestions into practice does not require any technical innovation, should be doable within a reasonable time frame and would substantially improve the manuscript. Besides these two major points, I only have a number of minor points (see below), all of which can be addressed editorially (unless the authors choose to take up the suggestion in minor point 14). If these points are addressed, I believe that the manuscript deserves publication in the EMBO Journal.

Major points

1. The authors did not detect a change in the degradation rate of Gas1*TMD when Pbn1 function was impaired by means of the PBN1-DAmP allele (Figure 2G, H). On the other hand, PBN1 overexpression accelerated Gas1*TMD degradation (Figure 2I, J). It is difficult to decide whether these observations indicate a direct role of Pbn1 in ERAD of non-GPI-APs. If

Pbn1 had such a role, the experiments may have failed to reveal it because the reduction of Pbn1 levels through the DAmP allele was not strong enough or because cells had adapted to it. I recommend to attempt to clarify this point by acutely depleting Pbn1 with the AID system (as in Figure 5) and looking at Gas1*^{TMD} degradation again. Similarly, I suggest to improve consistency across experiments by testing HA-Gas1* and HA-Gas1^{wt} degradation after auxin-induced Pbn1 depletion. Of the two systems used for Pbn1 depletion (DAmP allele and AID system), the AID is by far superior because of the acute rather than chronic disruption of Pbn1 function it affords. Hence, the paper would greatly benefit if this system were used for all key experiments.

We have now performed additional experiments using Gas1*, Gas1*^{TMD}, and Gas1^{WT} in combination with the acute Pbn1 depletion system based on the AID* degron. The resulting data, shown in Fig. 2O–U, provide an additional layer of evidence and corroborate our previous findings obtained with the DAmP allele of Pbn1.

We observed reduced molecular weights for Gas1* and Gas1*^{TMD}, but not for Gas1^{WT}, consistent with O-mannosylation deficiency in the former two constructs, which are misfolded.

No degradation defect was observed for Gas1^{wt}. As with cells expressing the DAmP allele of Pbn1, acute depletion of Pbn1 led to a degradation defect in Gas1* but not in Gas1*^{TMD} cells. This might reflect distinct conformations or molecular environments (GPI versus TM protein) that affect the readout of a mannosylated SRR, particularly when other strong degradation signals, such as modified N-glycans, are present.

2. To explore the role of O-mannosylation in ERAD, the authors compared the degradation of soluble CPY*, which is not O-mannosylated, and CPY*^{TMD}, which is O-mannosylated due to a serine-rich region. Pbn1 is then shown to have a role in ERAD of CPY*^{TMD} but not CPY* (Figure 3). However, CPY*^{TMD} differs from CPY* in two regards: the presence of the serine-rich region and also anchoring in the ER membrane, and it is not clear which factor is responsible for the different susceptibility to Pbn1-mediated ERAD. For a clean comparison, it is necessary to isolate the possible role of the serine-rich region, either by investigating a version of CPY*^{TMD} that does not possess the serine-rich region or by testing a version of soluble CPY* to which the serine-rich region has been added.

We have now generated a CPY*-based construct which lacks the TMD but retains the SRR (CPY*^{SRR}). Using this construct, we have performed experiments with the DAmP allele, with acute Pbn1 depletion and with Pbn1 overexpression. These data are now shown in Fig. 3L-S. With CPY*^{SRR} we observed increasing MW prior to degradation, consistent with protein O-mannosylation, and a dependency on Pbn1 for both O-mannosylation and degradation, linking the phenotypes to the presence of the SRR. Interestingly, a wild type version of HA-tagged CPY fused to the SRR does not show significant changes in MW and Pbn1 does not affect the slower turnover of this protein. Together these data support the model that the SRR can serve as a motif that is modified in a protein folding-dependent manner and contributes to ERAD when O-mannosylated.

Minor points

1. Abstract, line 53: "... through protein O-mannosylation is an essential component of ER quality control." The ERAD defects upon loss of Pbn1-Gpi14 function revealed in this study are rather subtle, so I think the word "essential" is too strong.

We have revised the Abstract and have added/modified text. We have removed "essential" and now have the following phrase: ..."Our results suggest that protein O-mannosylation constitutes a distinct glycan-dependent mechanism for regulating ERAD." (lines 54-55)

2. Introduction, line 65: "... retrotranslocation (...) followed by their ubiquitination and ..." This phrasing implies that proteins are retrotranslocated and only then ubiquitinated. Has this been established? To my understanding the order of events is not clear and ubiquitination may also occur during or even prior to retrotranslocation.

We agree that for specific substrates, such as for membrane proteins with cytosolic domains, ubiquitination may occur on these domains prior to protein retrotranslocation/membrane extraction. We have removed "followed by" to avoid misunderstanding and have now included the following phrase: "ERQC directs non-aggregated terminally misfolded polypeptides to ER-associated protein degradation (ERAD), which involves their retrotranslocation into the cytosol and ubiquitination by ER membrane-embedded protein complexes prior to delivery to the proteasome." (lines 60-63)

3. Introduction, line 93: "Drug-induced inhibition of protein N-glycosylation has been found to increase global protein O-mannosylation and reduce protein aggregation." As phrased, the sentence states that loss of N-glycosylation leads to less protein aggregation. Is this what the authors mean? Or do they mean that loss of N-glycosylation increases O-mannosylation and that O-mannosylation then helps to reduce protein aggregation?

Yes, loss of N-glycosylation increases O-mannosylation and that is thought to reduce protein aggregation. We have now introduced changes: "Drug-induced inhibition of protein N-glycosylation has been shown to enhance global protein O-mannosylation as a compensatory mechanism, which, in turn, has been proposed to mitigate protein aggregation." (lines 93-95)

4. Introduction, line 107: "... while a remaining pool ..." This should read 'the remaining pool'.

Thank you!

5. Results, line 181: "By combining genetics with enzyme treatment of cell lysates, we

confirmed ..." I think it would help the reader to spell out that "enzyme treatment" here means removal of glycans with PNGase F.

We have now included that information and the phrase now reads: "By combining genetics with treatment of cell lysates with the glycosidase PNGase F, ..." (lines 182-185)

6. Results, line 205: "In combination with the observed changes (...), it could suggest ..." I do not understand what 'it' refers to.

"It" referred to "these data" in the previous sentence. This phrase is no longer present in the revised manuscript since we have rewritten this section and included text associated with new data from experiments with the acute depletion of Pbn1. At the end of the section we have now included the following paragraph: "Together, these findings indicate that differential expression of Pbn1 specifically regulates the O-mannosylation and degradation of misfolded Gas1 variants, even those lacking a GPI anchor, suggesting additional cellular functions for Pbn1 beyond its established role in GPI anchor biosynthesis. In an appealing working model, Pbn1 function might directly or indirectly enhance O-mannosylation of misfolded proteins in order to promote their degradation. These results prompted further exploration of cellular Pbn1 functions." (lines 220-226)

7. Results, line 263 and elsewhere: The tagged Gpi14 protein is here referred to as Gpi14-tdimer but as Gpi14-tdimer2 elsewhere, for example in the legend of Figure 4. Please edit the manuscript to be consistent throughout.

Thank you. We have now corrected this inconsistency and have used *tdimer* consistently throughout the manuscript.

8. Results, line 309: The authors suggest that "(...) the observed connection of Pbn1 to protein O-mannosylation and ERAD includes Gpi14 and is important for a variety of misfolded proteins, including those that lack N-glycans and do not contain an SRR." Alternatively, Pbn1 and Gpi14 depletion may similarly impacted cell sensitivity to tunicamycin (Figure 5A) because GPI-AP biogenesis is impaired. Disrupted GPI-AP biogenesis could increase ER stress and thus further slow cell growth in the presence of tunicamycin. I therefore believe that the data are consistent with epistasis between Pbn1 and Gpi14 but that they do not allow conclusions about which of their shared functions (GPI-AP biogenesis or ERAD) are responsible for the observed growth phenotypes.

We agree with this concern. Reviewer #2 had similar concerns. We have now tuned down our conclusions, have moved the Figure to the EV section and have mentioned that these data are "consistent" with a contribution of Pbn1-Gpi14 to ER homeostasis: "Cells depleted of Pbn1 or Gpi14 exhibited increased sensitivity to tunicamycin, consistent with a role in maintaining ER homeostasis under N-glycosylation stress.(Fig. EV5F)." (lines 357-359)

9. Results, line 320: To help the reader, I suggest editing the text to "..., albeit less efficiently than CPY* (Figs. 5E and F, compare with Figs. 3A and B)."

We have now included the reference to Fig. 3A,B. (lines 332-334)

10. Results, line 334: I stumbled over the term "enrichment" because it was not clear to me relative to what the non-mannosylated form of Δ N-CPY* was enriched. Instead of enrichment, the term 'accumulation' might be simpler.

Due to the inclusion of new experimental data in this section, we have rewritten this paragraph and the corresponding phrase now reads: "Depletion of either component resulted in Δ Ng-CPY* stabilization, primarily of the non-mannosylated form of the protein (Fig. 5H-K)." (lines 348-350)

11. Results, line 345: "Additionally, Δ N-CPY* degradation was stabilized and ...". Shouldn't "degradation" be deleted here?

This phrase has been removed as part of the same section mentioned above and new text has been added.

12. Results, line 351: "... and are consistent with the observed sensitivity of cells to tunicamycin upon depletion of either Pgn1 or Gpi14 (Fig. 5A)." Since the data in Fig. 5A may be open to alternative interpretation (see minor point 8), it may be better to delete this part of the sentence and not muddy the conclusion drawn from Figure 5I - L.

We agree. As mentioned earlier, we have now moved the Figure to the EV section and rephrased the text. (lines 357-359).

13. Discussion, line 371: The authors state that their data suggest a role of O-mannosylation in both ERAD of N-glycosylated proteins and ERAD of proteins that lack other ERAD cues. The next paragraph begins with "Our findings related to SRRs support the first scenario", which makes it sound as if lines 371-373 had stated two mutually exclusive possibilities. Please edit for clarity, for instance by starting line 374 with something like: Our data provide evidence for both scenarios. First, our findings related to SRRs ...

We agree. We have changed the text accordingly: "Our findings support both scenarios." (line 423)

14. Discussion, line 393: The authors make the reasonable and testable suggestion that Yos9 may function in the recognition of ERAD cues based on O-mannosylation. It is a pity that they did not test this idea directly using ΔN -CPY*, and the reader is left wondering why this relatively simple experiment wasn't done. I therefore encourage the author to consider investigating the possible role of Yos9 already in this paper.

We fully comprehend this suggestion. We had briefly mentioned a hypothetical link to Yos9 in the Discussion section, as we lacked experimental data at the time. In the meantime, we have conducted experiments starting with the *yos9* deletion mutant. These experiments have generated a set of new and comprehensive data, which now forms the basis of ongoing research in our lab as a direct follow-up of this study. Since the data remain incomplete and, as we feel, would extend beyond the scope of this study, we have removed the paragraph that mentions Yos9 in the discussion section from the revised version—not because we consider the topic irrelevant, but in recognition of the reviewer's valid point.

15. Discussion, line 396: see minor point 13.

We have modified the text. (line 440)

16. Discussion, starting at line 406: I strongly agree with this paragraph, which provides a rationale to extend the use of the AID system to the experiments shown in Figure 2 (also see major point 1).

As mentioned earlier, we have performed and included these experiments in the revised version of the manuscript.

17. Discussion, line 449: The discussion ends rather abruptly, and a brief summary paragraph would be nice.

We have significantly revised the Discussion section in response to reviewer comments, new experimental data and in order to enhance overall stringency. In addition, a brief summary paragraph has been added (see also minor point 18). (lines 490-499)

18. Discussion: The data convincingly support a role for Pbn1/Gpi14-mediated O-mannosylation in ERAD, and the authors appear to favour the idea that O-glycosylation acts as ERAD "cues". However, is it possible that this O-mannosylation acts to end futile folding cycles as proposed by Xu et al., 2013? Unless this idea can be discarded for some obvious reason, the authors should bring it up as an alternative model to lectin-mediated recognition of O-mannosylation-based degrons.

We have taken this point into account and have replaced strong and affirmative statements about O-mannoses being bona-fide ERAD “cues” with terms that are more inclusive. For instance, we have removed the term “ERAD cues” from the Abstract and replaced it with “...thereby promoting ERAD.” (line 52). We now discuss this point in the summary paragraph (also in connection to minor point 17).

19. Overall, the paper is quite concise overall with just five figures. The authors could therefore consider integrating some of supplemental data into main figures to make them more visible. This, however, is just an idea the authors can feel free to ignore.

For the revised version, we have included additional data and now present six figures, some of which include data previously found in supplementary figures, for example Fig. 5L,M, whereas other data have been moved from the main figure to an expanded view (EV) figure, for example Fig. EV5F. All our data are now presented in six main figures and six EV figures, whereas Table EV1 shows our screen data.

Dear Veit,

We have now received re-review reports from two of the three referees, which I have included below. As you will see, you have addressed their concerns satisfactorily. As the third referee has not yet responded, I am prepared to move forward towards publication without this input on the proviso that no substantive concerns are raised from Referee #2. Please, though, address the small point of clarification raised by Referee #3. Before I can finally accept the manuscript, there are some remaining editorial points which need to be addressed. In this regard would you please:

- acknowledge the following funding from the Blythe Brenden-Mann Foundation; the "Talento Doctor" Fellowship co-funded by the European Regional Development Fund (ERDF) and the Junta de Andalucía, Spain in our online submission system, change the title of the conflict of interest statement to the "Disclosure and competing interests statement",
- update source file names, titles, legends and manuscript callouts to Dataset EV1 instead of Table EV1; this should be uploaded individually as a Dataset file with legend in a separate tab/sheet in the Excel file instead of a README file,
- save source data files in a scheme one figure/folder and then uploaded as .zip files. E.g. all the Source data files for figure 1 need to be saved in a single folder and this needs to be zipped and then uploaded as "SD figure 1.zip" file. For EV and/or appendix figures, then ZIP together all source data,
- provide exact p values in the legends of figures EV2 F, EV3 B, D, F,
- define box plots in terms of minima, maxima, centre, bounds of box and whiskers, and percentile in the legend of figure EV3 F,
- define the measure of centre for error bars in the legends of figures 2D, F, H, J, L, N, Q, S, U; 3G, I, K, M, O, Q; 5E, I, K, M; 6J, EV2 F; EV3 B, D; EV5 C, E,
- remove the "Abbreviations" section, and
- use the following section order: Title page - Abstract - Keywords - Introduction - Results - Discussion - Methods - Data Availability - Acknowledgements - Disclosure and Competing Interests Statement - References - Figure Legends - Table(s) - Expanded View Figure Legends.

We include a synopsis of the paper (see <http://emboj.embopress.org/>). Please provide me with a general summary image, a two sentence statement and 3-5 bullet points that capture the key findings of the paper.

I am looking forward to receiving your revised manuscript.

EMBO Press is an editorially independent publishing platform for the development of EMBO scientific publications.

Best wishes,

William

William Teale, PhD
Editor
The EMBO Journal
w.teale@embojournal.org

We realize that it is difficult to revise to a specific deadline. In the interest of protecting the conceptual advance provided by the work, we recommend a revision within 3 months (14th Jan 2026). Please discuss the revision progress ahead of this time with the editor if you require more time to complete the revisions. Use the link below to submit your revision:

Referee #1:

My concerns on the evidence that this is a direct effect of O-mannosylation of the misfolded protein substrates and other concerns have been addressed. The manuscript is substantially improved.

Referee #3:

Review of revised version

The authors have responded adequately to my previous comments, in particular by adding new data to Figure 2 and 3. These data strengthen their conclusions and alleviate the concerns I had earlier.

The only minor comment I have here is that it is difficult for the reader to understand why the authors used the soluble CPY* in Figure 3A-D, then switched to a transmembrane version of CPY* in Figure 3E-K, and finally return to soluble versions of CPY* in Figure 3L-S. This back and forth is disorienting. Perhaps the authors could shift the results with transmembrane CPY* to the end of the figure so that all the data with soluble CPY* are grouped together. In any case, the authors should give an explanation in the text why they used a transmembrane version at all (line 234 would be the place to do it).

Overall, I think this is a now very nice paper, I congratulate the authors on their work and support publication in the EMBO Journal.

Point-by-point response

REFEREE #3:

The only minor comment I have here is that it is difficult for the reader to understand why the authors used the soluble CPY* in Figure 3A-D, then switched to a transmembrane version of CPY* in Figure 3E-K, and finally return to soluble versions of CPY* in Figure 3L-S. This back and forth is disorienting. Perhaps the authors could shift the results with transmembrane CPY* to the end of the figure so that all the data with soluble CPY* are grouped together. In any case, the authors should give an explanation in the text why they used a transmembrane version at all (line 234 would be the place to do it).

Major concerns:

In our revised manuscript, we retained the original data obtained with CPY*SRR_TMD in Figure 3, even after adding new data from the requested experiments using CPY*SRR and CPYwtSRR constructs, as we were not instructed to remove it. In fact, all data combined underscore the role of the SRR in degradation of misfolded species and the current order of data also best illustrates our iterative approach used in this series of experiments. Although the reviewer is right in noticing that we use both soluble and membrane integrated constructs, all experiments with a particular construct are grouped together and there is no repetition. Construct CPY* (Figures 3A-D): this construct concerns a standard ERAD model. Construct CPY*SRR_TMD (Figures 3E-K): this construct is the simplest chimeric construct for a bold approach to addressing the role of the SRR. CPY*SRR (Figure 3L-Q): this construct is a more specific construct for unequivocally addressing the role of the SRR. CPYwtSRR (Figure 3R-S): this construct serves as a control.

Following the recommendations of reviewer #3 we have now slightly modified the text, also to better explain the use of CPY*SRR_TMD and CPY*SRR and to illustrate better our iterative approach:

“By simply fusing the C-terminal domain of Gas1*TMD, which includes its TMD and part of its proximal serine-rich region (SRR), to the C-terminus of CPY* we generated the diagnostic construct CPY*SRR_TMD (Fig. 3E, SRR in yellow).” (lines 224-26).

and

“To distinguish potential effects of the TMD on the degradation properties of CPY*SRR_TMD, we next generated CPY*SRR, which lacks the TMD.” (lines 251-52).

Dear Veit,

I am pleased to inform you that your manuscript has been accepted for publication in the EMBO Journal.

Congratulations on a really nice study!

Best wishes,

William

William Teale, PhD
Editor
The EMBO Journal
w.teale@embojournal.org

Please note that it is The EMBO Journal policy for the transcript of the editorial process (containing referee reports and your response letters) to be published as an online supplement to each paper. If you should prefer removal of any referee-only figures included in the point-by-point response(s), e.g. because they may still be used for future publication or because they have been reproduced from published work by others, please do let us know immediately via response email.

More information is available here: https://www.embopress.org/transparent-process#Review_Process
